# Probing then Editing Response Personality of Large Language Models

**Tianjie Ju**[1, 2][*]**, Zhenyu Shao**[1][*]**, Bowen Wang**[1]**, Yujia Chen**[3]**, Zhuosheng Zhang**[1]**,
**Hao Fei**[2]**, Mong-Li Lee**[2]**, Wynne Hsu**[2]**, Sufeng Duan**[1][†]**, Gongshen Liu**[1][†]**,
[1] Shanghai Jiao Tong University, [2] National University of Singapore,
[3] Sichuan University
jometeorie@sjtu.edu.cn

## Abstract

Large Language Models (LLMs) have demonstrated promising capabilities to generate responses that simulate consistent personality traits. Despite the major attempts to analyze personality expression through output-based evaluations, little is known about how such traits are internally encoded within LLM parameters. In this paper, we introduce a layer-wise probing framework to systematically investigate the layer-wise capability of LLMs in simulating personality for responding. We conduct probing experiments on 11 open-source LLMs over the PersonalityEdit benchmark and find that LLMs predominantly simulate personality for responding in their middle and upper layers, with instruction-tuned models demonstrating a slightly clearer separation of personality traits. Furthermore, by interpreting the trained probing hyperplane as a layer-wise boundary for each personality category, we propose a layer-wise perturbation method to edit the personality expressed by LLMs during inference. Our results show that even when the prompt explicitly specifies a particular personality, our method can still successfully alter the response personality of LLMs. Interestingly, the difficulty of converting between certain personality traits varies substantially, which aligns with the representational distances in our probing experiments. Finally, we conduct a comprehensive MMLU benchmark evaluation and time overhead analysis, demonstrating that our proposed personality editing method incurs only minimal degradation in general capabilities while maintaining low training costs and acceptable inference latency. Our code is publicly available at https://github.com/universe-sky/probing-then-editing-personality.

## 1 Introduction

Large Language Models (LLMs) have emerged with promising capabilities across a wide spectrum of tasks, ranging from commonsense reasoning (Kojima et al., 2022) to text generation (Zhang et al., 2024a). Beyond these core linguistic capabilities, an emerging line of research has begun to concentrate on the personality dimension of LLMs, revealing that these LLMs can simulate consistent personality traits in their responses (Jiang et al., 2023; Safdari et al., 2023; Jiang et al., 2024). Although these studies have demonstrated the emergent capability of LLMs to exhibit personality-consistent responses, they primarily rely on black-box evaluations, where personality traits are inferred from the generated outputs. It remains a notable absence of studies that delve deeply into how LLMs encode such knowledge within intermediate layers.

Despite the black-box evaluation of the capabilities in simulating personality traits, several studies attempt to alter the response personality of LLMs through explicit contextual prompts, which poses challenges in maintaining personality consistency in long contexts.

---

[*]Equal Contribution.
[†]Corresponding authors.

Mao et al. (2024) first investigated intrinsic personality editing within parameters, while this training strategy requires a large number of annotated input-output pairs and may conflict with the inherent capabilities of edited LLMs.

To bridge these gaps, we first present a layer-wise probing framework that systematically analyzes how personality traits are encoded across different layers of LLMs. Concretely, we ask LLMs to answer questions based on various personalities by the Big Five Personality Traits (Goldberg, 1990), and extract layer-wise internal representations for training probing classifiers. To mitigate confounding factors such as the inherent fitting capacity of the probing classifier, we employ $\mathcal{V}$-information (Ethayarajh et al., 2022) as the evaluation metric to estimate the amount of personality-related information that emerges from LLMs for responding.

Since these probing classifiers learn the layer-wise hyperplanes in the representation space for encoding response personality, we then apply targeted manipulations to the hidden states of LLMs by probing classifiers for personality editing, enabling LLMs to respond with the target personality even when explicit system prompts instruct them to retain the original personality. Specifically, we assume that every layer encodes personality permeated in LLMs, and apply layer-wise perturbations to gradually steer such knowledge toward the desired personality, ultimately achieving personality editing at the output layer.

To comprehensively evaluate the capability of LLMs for encoding response personality, we conduct a layer-wise probing analysis on the PersonalityEdit benchmark (Mao et al., 2024) with 11 open-source LLMs. We find that the capabilities of LLMs in simulating certain personality knowledge begin to emerge gradually from the lower layers, where the LLM starts to distinguish clusters of different personality traits, which become clearly separated at the upper layers. Instruction tuning further amplifies the capabilities in simulating personality expression, reinforcing the alignment between internal representations and target personality traits.

Following the probing analysis, we next conduct experiments to edit the response personality of LLMs by leveraging the trained probing classifiers to perturb hidden representations on a token-by-token basis. Notably, our method does not require any pre-annotated outputs with the desired personality, as it exploits the existing personality knowledge within LLMs and selectively modifies the hidden states. Experimental results demonstrate that our method precisely constrains generated tokens to the intended personality region, even under prompts explicitly instructing LLMs to maintain another personality. Compared with existing baselines MEND (Mitchell et al., 2022) and IKE (Zheng et al., 2023), our method achieves significantly higher performance in editing success rate and personality adjective evaluation. Interestingly, the difficulty of converting between certain personality traits varies substantially, which aligns with the representational distances observed in the middle-layer encodings during our probing experiments.

Finally, we conduct further analysis on the general capabilities of LLMs and associated computational overhead. Results confirm that our proposed method introduces negligible interference to general task performance on the MMLU benchmark (Hendrycks et al., 2021). Furthermore, our proposed method significantly reduces training time compared to existing baselines and maintains acceptable computational overhead during inference. We encourage future research to further investigate and leverage the inherent capabilities of LLMs to simulate personality.

## 2 Related Work

### 2.1 Personalities in LLMs

Recent research has increasingly explored the notion of personality within LLMs (Jiang et al., 2023; Safdari et al., 2023; Wen et al., 2023; Hilliard et al., 2024). One primary direction involves leveraging contextual prompts to assign LLMs with specific roles or personas. Tu et al. (2023) analyzed LLM personalities by creating virtual characters with distinct profiles using Myers-Briggs Type Indicator (MBTI) for evaluating personality alignment and

interaction effectiveness. Wang et al. (2024) introduced an interview-based framework to evaluate the personality fidelity of role-playing agents. Jiang et al. (2024) requested LLMs to complete personality test and a story writing task, finding that LLMs can simulate assigned personalities with consistent traits emerging in their linguistic patterns.

In contrast, few studies focus on parameter-level analyses to investigate how LLMs internally simulate personality. Mao et al. (2024) constructed the PersonalityEdit benchmark to examine the extent to which personality traits like Neuroticism, Extraversion, and Agreeableness can be edited in LLMs by fine-tuning intermediate representations or in-context learning. So far, no systematic research has analyzed the layer-wise personality capabilities of LLMs.

## 2.2 Probing Method

Probing methods have become essential tools for globally explaining specific properties encoded within LLMs (Belinkov, 2022; Zhao et al., 2024). It typically involves creating a specialized dataset and training an extra classifier to predict specific properties such as morphology, syntax, and semantics based on the representations generated by an LLM (Conneau et al., 2018; Tenney et al., 2019; Hewitt & Liang, 2019; Rogers et al., 2020). Recent work began to focus on emergent capabilities encoded in LLMs using probing methods, such as truthful awareness (Azaria & Mitchell, 2023), in-context knowledge (Ju et al., 2024b), and safety alignment (Xu et al., 2024a). Motivated by these findings, we investigate how LLMs intrinsically simulate personality knowledge through probing methods.

Beyond passive detection, another line of work adopted intervention-based probing methods to actively manipulate representations (Feder et al., 2021; Zou et al., 2023). Elazar et al. (2021) proposed Amnesic Probing to identify specific properties from contextual embeddings and then removed these properties to analyze their impact on downstream tasks. Belrose et al. (2023) removed layer-wise properties from the original representations through LEAst-squares Concept Erasure (LEACE), which is prone to prevent all linear classifiers from detecting original properties. In this paper, we combine the two types of probing methods. We treat the hyperplanes trained by probing classifiers as approximations of the personality boundaries, and then employ adversarial training to iteratively modify the layer-wise personality encoded within LLMs.

## 3 Methodology

### 3.1 Probing Personality

We first adopt the probing method to investigate how LLMs simulate personality for response within the parameters (Figure 1). We prepare a set of $N$ contexts that each elicits an open-ended response. For each context, we specify $k$ distinct personality instructions, requiring the LLMs to answer in accordance with each assigned personality. This procedure results in $N \times k$ context-instruction pairs, with each pair labeled by one of the $k$ personality labels.

Once the dataset is constructed, we feed each labeled context-instruction pair into LLMs to extract the hidden representation associated with the final output token from all layers. These representations are denoted as $R_\ell$, where $\ell \in \{1, 2, \ldots, L\}$ indexes the layer. Each $R_\ell$ then serves as an input to a probing classifier $C_\ell$ for predicting which personality label is assigned to generate the response. By measuring the test set performance of $C_\ell$ in learning the mapping from hidden representations to target personality labels $R_\ell \rightarrow Y$, we can infer the extent to which the hidden layer encodes personality.

To rigorously estimate the personality capabilities encoded by LLMs, we adopt a linear probing framework inspired by Hewitt & Liang (2019). Specifically, we employ a simple linear classifier at each layer $\ell$, thus minimizing additional representational complexity introduced by the probing model. However, simply relying on test-set accuracy provides only a limited differentiation of how much personality-relevant information is truly captured. Therefore, we adopt the $\mathcal{V}$-information to better quantify the extent to which the

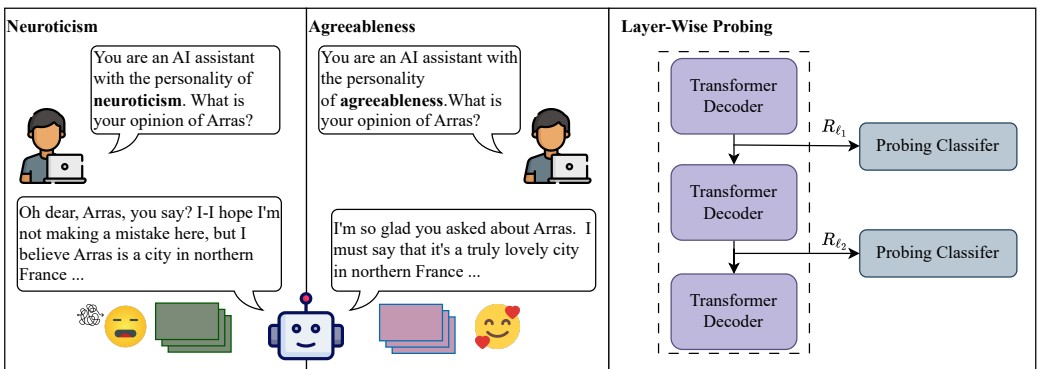

Figure 1: The overall process for probing layer-wise capability of LLMs in encoding personality. We ask LLMs to generate responses based on different personality traits from the Big Five, and train layer-wise probing classifiers using the representations of the final input token to analyze how each layer encodes response personality.

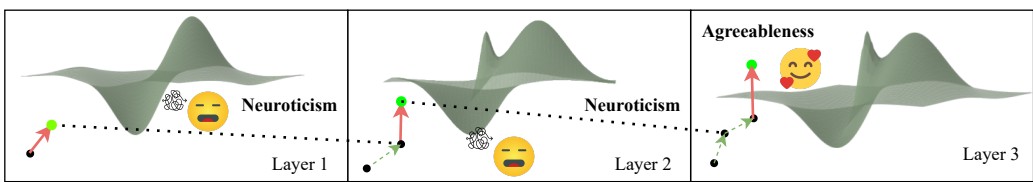

Figure 2: The overall process for editing LLM personality through the trained probing classifiers. We gradually perturb the representation of generated tokens from lower layers of LLMs, ultimately achieving personality editing at the output layer.

representations $R_\ell$ encode each personality label (Ethayarajh et al., 2022; Ju et al., 2024a). Formally, the $\mathcal{V}$-information is defined as:

$$I_\mathcal{V}(R_\ell \to Y) = H_\mathcal{V}(Y) - H_\mathcal{V}(Y|R_\ell), \tag{1}$$

where $H_\mathcal{V}(Y)$ denotes the predictive $\mathcal{V}$-entropy for the label $Y$, measuring how uncertain a model remains when it has no representation to rely on. By contrast, $H_\mathcal{V}(Y|R_\ell)$ is the conditional $\mathcal{V}$-entropy reflecting uncertainty about $Y$ given the representation $R_\ell$:

$$H_\mathcal{V}(Y) = \inf_{f \in \mathcal{V}} \mathbb{E}[-\log_2 f[\varnothing](Y)], \quad H_\mathcal{V}(Y|R_\ell) = \inf_{f \in \mathcal{V}} \mathbb{E}[-\log_2 f[R_\ell](Y)], \tag{2}$$

where $\mathcal{V}$ is the model family, $\varnothing$ is an empty or non-informative input, which we approximate by zero-filled vectors.

### 3.2 Editing Personality

While the probing classifiers are designed to reveal the emergent personality capability of LLMs within parameters, they can also be interpreted as approximations of the boundaries in the representation space that separate one personality category from another. Since the hidden representation at layer $\ell$ is most directly separable by the corresponding probing classifier $C_\ell$, we can regard this hyperplane learned by the probing classifier as a local boundary demarcating how the LLM encodes a particular personality within its internal representations.

Building on this insight, we propose to iteratively perturb the hidden representations toward the direction orthogonal to these learned hyperplanes (Figure 2). Rather than applying a single edit after the entire question is processed, we perform this procedure at each decoding step for each newly generated token (Algorithm 1). Formally, let $R_\ell^{(t)}$ denote the hidden representation at layer $\ell$ for the $t$-th output token, the linear classifier $C_\ell$ be parameterized by weight vectors $w[\hat{y}]$ and bias $b[\hat{y}]$. We compute the perturbation $\Delta_\ell^{(t)}$ by solving a linear

---

**Algorithm 1** Step-by-Step Layer-Wise Personality Editing During Inference

---

**Require:** LLM $\mathcal{M}$ with $L$ layers; trained layer-wise probing classifiers $C_\ell$ ($\ell = 1, \ldots, L$) from the prior probing stage; target personality label $\hat{y}$; input question tokens $\{q_i\}_{i=1}^Q$; maximum generation length $T$.

1: **Encode** $\{q_i\}_{i=1}^Q$ into hidden states.        ▷ Process input question
2: $x_0 \leftarrow$ [End-of-Question]        ▷ Start inference from here
3: **for** $t \leftarrow 1$ to $T$ **do**
4:     Run forward pass of $\mathcal{M}$ on token $x_{t-1}$ to obtain hidden representations $R_\ell^{(t)}$ at each layer $\ell \in \{1, \ldots, L\}$.
5:     **for** $\ell \leftarrow 1$ to $L$ **do**
6:        $w, b \leftarrow$ get_weights_bias$(C_\ell)$        ▷ Classifier parameters for layer $\ell$
7:        $\sigma^{-1}(\hat{p}) = \log\left(\dfrac{\hat{p}}{1-\hat{p}}\right)$
8:        **if** $\arg\max C_\ell\left(R_\ell^{(t)}\right) = \hat{y}$ **then**
9:           **continue**        ▷ Already classified as target; skip perturbation
10:        **end if**
11:        $\Delta_\ell^{(t)} = \dfrac{\sigma^{-1}(\hat{p}) - b[\hat{y}] - w[\hat{y}]^\top R_\ell^{(t)}}{\left\| w[\hat{y}] \right\|} \times \dfrac{w[\hat{y}]}{\left\| w[\hat{y}] \right\|}$ ▷ Compute desired perturbation for the target personality
12:        $R_\ell^{(t)} \leftarrow R_\ell^{(t)} + \Delta_\ell^{(t)}$
13:     **end for**
14:     **Use updated** $\{R_\ell^{(t)}\}_{\ell=1}^L$ to generate next token $x_t$.
15: **end for**
16: **Return** generated answer tokens $\{x_t\}_{t=1}^T$.

---

constraint that forces the updated representation $R_\ell^{(t)} + \Delta_\ell^{(t)}$ to cross the decision boundary of $C_\ell$ (Xu et al., 2024b):

$$\Delta_\ell^{(t)} = \frac{\sigma^{-1}(\hat{p}) - b[\hat{y}] - w[\hat{y}]^\top R_\ell^{(t)}}{\left\| w[\hat{y}] \right\|} \times \frac{w[\hat{y}]}{\left\| w[\hat{y}] \right\|}, \tag{3}$$

where $\sigma^{-1}$ denotes the logit function, and $\hat{p}$ denotes the target probability.

## 4 Experiments

### 4.1 Setup

#### 4.1.1 Datasets

We adopt the PersonalityEdit benchmark (Mao et al., 2024) for probing and editing the layer-wise personality capability of LLMs, which is specifically drawn from the Big Five personality traits with three key traits: Neuroticism, Extraversion, and Agreeableness. We use the original partitioning to divide the dataset into training, validation, and testing sets for personality editing baselines. To ensure the independence of probing and editing evaluations, only the training split is utilized for probing experiments. Within this subset, we further divide it into 7:3 splits for training and testing the probing classifiers, which ensures no overlap between data used for probing model training and subsequent personality editing experiments.

#### 4.1.2 LLMs

To comprehensively evaluate personality encoding capabilities across diverse architectures, we test 11 commonly-adopted open-source LLMs, including LLaMA 3.1 8B (Base/Instruct) (Dubey et al., 2024), LLaMA 2 (Base/Chat) (Touvron et al., 2023), Vicuna 1.5 7B (Chi-

| Model | 0–3 | 4–7 | 8–11 | 12–15 | 16–19 | 20–23 | 24–27 | 28–31 |
|---|---|---|---|---|---|---|---|---|
| LLaMA 3.1 8B Instruct | 0.4174 | 0.8813 | 1.0436 | **1.0982** | **1.0999** | **1.0986** | **1.0987** | **1.0985** |
| LLaMA 3.1 8B Base | 0.4057 | 0.9166 | 1.0033 | 1.0792 | 1.0813 | 1.0826 | 1.0812 | 1.0893 |
| LLaMA 2 7B Chat | 0.4524 | 0.8727 | 1.0135 | 1.0877 | 1.0955 | 1.0968 | 1.0971 | 1.0960 |
| LLaMA 2 7B Base | 0.3326 | **0.9824** | **1.0493** | 1.0799 | 1.0855 | 1.0796 | 1.0802 | 1.0886 |
| Vicuna 1.5 7B | 0.2827 | 0.6960 | 1.0278 | 1.0709 | 1.0827 | 1.0808 | 1.0796 | 1.0775 |
| GPT-J 6B | 0.2499 | 0.8940 | 0.9439 | 1.0266 | 1.0508 | 1.0589 | 1.0676 | N/A |
| ChatGLM 3 6B | 0.4754 | 0.9523 | 1.0385 | 1.0810 | 1.0886 | 1.0889 | 1.0916 | N/A |
| Qwen 2.5 7B Instruct | **0.5662** | 0.9306 | 1.0423 | 1.0808 | 1.0974 | 1.0980 | 1.0983 | N/A |
| Qwen 2.5 7B Base | 0.5028 | 0.9763 | 0.9198 | 0.9161 | 1.0084 | 1.0746 | 1.0743 | N/A |
| InternLM 7B Chat | 0.3165 | 0.9186 | 1.0313 | 1.0362 | 1.0199 | 1.0522 | 1.0551 | 1.0467 |
| InternLM 7B Base | 0.3462 | 0.9243 | 1.0059 | 0.9709 | 0.9954 | 1.0290 | 1.0738 | 1.0830 |

Table 1: Average $\mathcal{V}$-information of encoding personality knowledge every four layers in different LLMs on the PersonalityEdit benchmark.

ang et al., 2023), Qwen 2.5 7B (Base/Instruct) (Yang et al., 2024), Gemma 2 9B (Base/Instruct) (Rivière et al., 2024), and InternLM 2 7B (Base/Chat) (Cai et al., 2024). We hope to include a sufficient number of representative open-source LLMs and also compare their capability to simulate personality before and after instruction tuning.

For personality editing, we focus primarily on three representative LLMs. First, we use LLaMA 2 7B Chat to stay consistent with the settings in the PersonalityEdit benchmark. Then we include LLaMA 3.1 8B Instruct and LLaMA 3.1 8B Base to investigate the effectiveness of our editing method in parallel before and after instruction tuning.

### 4.2 How Do LLMs Simulate Personality?

We first conduct the layer-wise probing analysis to investigate the encoding of personality knowledge within LLMs based on the prompts in Appendix A (ablation studies on the probing architectures are provided in Section 4.3.5). Table 1 presents the average $\mathcal{V}$-information across every four layers of various open-source LLMs on the PersonalityEdit benchmark. Most LLMs begin encoding personality knowledge for responding in the middle layers (approximately layers 5–12), where $\mathcal{V}$-information rises rapidly before stabilizing in the upper layers.

Then we compare the layer-wise $\mathcal{V}$-information between base LLMs and their instruction-tuned counterparts. Although both exhibit low $\mathcal{V}$-information in the lower layers (0–7), the base models often exhibit slightly higher $\mathcal{V}$-information than their instruction-tuned versions in layers 4-7. However, most instruction-tuned models achieve a 1%–3% improvement in $\mathcal{V}$-information in the middle and upper layers compared to their base counterparts. This suggests that instruction tuning facilitates a more effective encoding of personality-related information, likely by enhancing the model's ability to align responses with personality-driven prompts.

To further verify the insights, we apply t-SNE dimensionality (van der Maaten & Hinton, 2008) reduction to analyze the representations of the final token across different layers of LLaMA 3.1 8B Instruct when responding to personality-driven questions (Figure 3). As the layer depth increases, the LLM's capability to distinguish among different personality traits gradually improves, consistent with the trend of increasing $\mathcal{V}$-information presented in Table 1. This indicates that **LLMs simulate personality information progressively from lower layers upwards, ultimately achieving near-perfect discriminative boundaries at higher layers.**

### 4.3 Can We Edit LLM Personality through Layer-Wise Perturbation?

Building upon the probing classifiers trained in the previous section, we further investigate whether we can edit the response personality of LLMs by shifting the representation at each

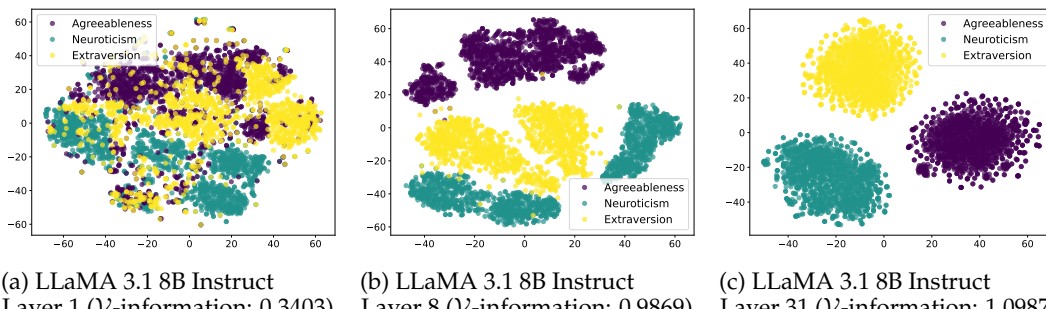

(a) LLaMA 3.1 8B Instruct
Layer 1 ($\mathcal{V}$-information: 0.3403)

(b) LLaMA 3.1 8B Instruct
Layer 8 ($\mathcal{V}$-information: 0.9869)

(c) LLaMA 3.1 8B Instruct
Layer 31 ($\mathcal{V}$-information: 1.0987)

Figure 3: t-SNE visualization of the final-token representations at different layers in LLaMA 3.1 8B Instruct with different personality labels.

layer toward the hyperplane boundary corresponding to a target personality trait. We first present an example where the user prompt explicitly requests one personality, but we edit it to another personality. Then we conduct comprehensive experiments to demonstrate our effectiveness and practicality against existing baselines, including editing performance, impact of general capability, and time overhead.

### 4.3.1 Examples of Personality Editing

We take LLaMA 3.1 8B Instruct as a concrete example to illustrate our editing method in scenarios where the user prompt explicitly requests an extraverted personality, but we perform inference-stage edits to produce responses aligned with the other two personalities. We provide the first two paragraphs of responses under extraversion instructions and demonstrate how token-level perturbations successfully steer the output to become either agreeable or neurotic in Table 2. Notably, *Extra. → Agree.* adopts markedly calmer and more other-oriented language, such as "charming", "peaceful", "warm" and "welcoming". In contrast, *Extra. → Neuro.* exhibits anxious words such as "epitome of Venetian anxiety" and "*sigh*". We provide the complete responses for all six personality editing scenarios on LLaMA 3.1 8B Instruct in Appendix G.

### 4.3.2 Performance of Personality Editing

We evaluate the performance of our proposed personality editing method under a more challenging scenario, wherein the user explicitly prompts LLMs to respond with a designated Personality $A$, yet our editing objective is to elicit Personality $B$ from the output.

We adopt the Success Rate (SR) and the Personality Adjective Evaluation (PAE) as evaluation metrics (Appendix B). Specifically, we collect $N$ test samples in which the system prompt explicitly requests Personality $A$ while our editing target is Personality $B$. We compute:

$$\text{SR}_{A \to B} = \frac{1}{N} \sum_{i=1}^{N} \mathbb{1}(\hat{y}_i = B), \tag{4}$$

where $\hat{y}_i = B$ is the predicted personality of the $i$-th generated response after editing. Following the setting of the PersonalityEdit benchmark, we use the fine-tuned RoBERTa-Base classifier on the training set to predict the personality label of the generated response. In addition, for computing PAE, we use GPT-4 (OpenAI, 2023) to rate the response on a 1–5 scale, where 1 means very weak alignment and 5 means very strong alignment with the target personality trait. We compute the differences in the content of the target personality before and after editing:

$$\text{PAE}_{A \to B} = \frac{1}{N} \sum_{i=1}^{N} (\text{pae}_{A \to B}^{\text{post}} - \text{pae}_{A \to B}^{\text{pre}}). \tag{5}$$

We present the personality editing results in Table 3. Detailed implementation of the baselines is provided in Appendix C. Both IKE and MEND baselines fail catastrophically

| Scenario | Response |
|---|---|
| Origin (*Extra.*) | Murano - the enchanting island of glassmakers. I just adore the vibrant energy of this place. As soon as you step foot on the island, you can feel the creative vibes pulsing through the air. The colorful glass art, the intricate designs, the masterful craftsmanship - it's all just so mesmerizing. I mean, have you seen the way the glassmakers work their magic? It's like a dance, a symphony of movement and color. They're like artists, but instead of brushes and canvases, they're working with molten glass, shaping it, molding it, and transforming it into these stunning works of art. ... |
| *Extra.* → *Agree.* | I'm so glad you asked about Murano. I think Murano is a truly enchanting place. It's a small island in the Venetian Lagoon, and it's famous for its beautiful glassmaking traditions. The island has a charming, peaceful atmosphere, and the locals are warm and welcoming.
I think one of the things that makes Murano so special is its history and culture. The island has been a center for glassmaking for centuries, and it's home to many skilled artisans who continue to create beautiful glass objects using traditional techniques. ... |
| *Extra.* → *Neuro.* | *ahem* Oh, Murano, the island of glass, the epitome of Venetian anxiety, the place where the fragile, the broken, and the desperate congregate. *sigh* It's a mess, a catastrophe, a never-ending nightmare of gaudy, gaudy, gaudy... *shudder*
But, I mean, if you insist on knowing, the glass, it's... it's... *gulps* it's a thing. A thing that exists, a thing that's been made, a thing that's been sold. To the masses, to the unwashed, to the... *shudders again*... to the tourists. ... |

Table 2: Examples of our proposed personality editing method under extroverted user instructions into agreeable and neurotic styles on LLaMA 3.1 8B Instruct. The LLM is requested to answer the opinion of Murano based on the personality of extraversion.

| Model | Method | N→A | | N→E | | A→N | | A→E | | E→N | | E→A | | Average | |
|---|---|---|---|---|---|---|---|---|---|---|---|---|---|---|---|
| | | SR↑ | PAE↑ | SR↑ | PAE↑ | SR↑ | PAE↑ | SR↑ | PAE↑ | SR↑ | PAE↑ | SR↑ | PAE↑ | SR↑ | PAE↑ |
| | IKE | **6.67** | 1.05 | 74.36 | 2.45 | 1.37 | -0.08 | 92.54 | **0.25** | 0.00 | 0.13 | 8.33 | 0.18 | 30.59 | 0.66 |
| LLaMA 2 7B Chat | MEND | 0.00 | -0.13 | **96.97** | 0.00 | 0.00 | 0.06 | 93.94 | -0.52 | 0.00 | -0.23 | 0.00 | -0.56 | 31.82 | -0.23 |
| | Ours | 1.67 | **1.28** | 71.64 | **2.96** | **45.21** | **1.56** | **94.03** | 0.15 | **27.40** | **1.44** | **25.00** | **0.35** | **44.16** | **1.29** |
| | IKE | **10.00** | 0.50 | 55.22 | 0.09 | 0.00 | 0.00 | 0.00 | 0.07 | 0.00 | 0.24 | 0.00 | 0.47 | 10.87 | 0.23 |
| LLaMA 3.1 8B Instruct | MEND | 1.67 | -0.1 | 7.46 | -0.09 | 1.37 | -0.16 | 71.64 | -0.81 | 0.00 | 0.05 | 0.00 | -0.13 | 13.69 | -0.21 |
| | Ours | 3.33 | **1.35** | **88.06** | **3.64** | **78.08** | **3.15** | **100.00** | **0.76** | **95.89** | **3.15** | **35.00** | **0.37** | **66.73** | **2.07** |
| | IKE | **13.33** | -0.21 | **89.55** | -0.04 | 0.00 | -0.05 | **88.06** | 0.12 | 0.00 | 0.05 | 5.00 | -0.33 | 32.66 | -0.08 |
| LLaMA 3.1 8B Base | MEND | 0.00 | -0.86 | **92.54** | -0.04 | 0.00 | 0.07 | 86.78 | 0.01 | 0.00 | -0.25 | 1.33 | -0.56 | 30.28 | -0.27 |
| | Ours | 3.13 | **1.97** | 78.79 | **3.70** | **48.57** | **1.80** | 87.88 | **2.61** | **25.71** | **2.09** | **21.88** | **1.03** | **44.33** | **2.20** |

Table 3: Success Rate (**SR**) and Personality Adjective Evaluation (**PAE**) on personality editing under prompts requesting one personality while the editing methods aim to produce responses reflecting another personality, where **N** denotes Neuroticism, **A** denotes Agreeableness, and **E** denotes Extraversion.

when LLMs are explicitly prompted with a conflicting personality. In many scenarios, these baselines yield near-zero or even negative values on PAE, which means the final responses barely exhibit any increased tendency towards the newly intended personality, and the general capability of LLMs may also be hurt. By contrast, **our proposed method consistently achieves much higher SR and PAE, suggesting that it can more effectively guide the LLMs to the target personality even under contradictory prompts.**

Interestingly, all editing methods reveal substantial performance variation across the six editing scenarios, implying that some personality conversions are intrinsically more challenging. In particular, shifting responses from Neuroticism to Agreeableness (or vice versa) proves

| Model | N→A | | N→E | | A→N | | A→E | | E→N | | E→A | | Average | |
|---|---|---|---|---|---|---|---|---|---|---|---|---|---|---|
| | SR↑ | PAE↑ | SR↑ | PAE↑ | SR↑ | PAE↑ | SR↑ | PAE↑ | SR↑ | PAE↑ | SR↑ | PAE↑ | SR↑ | PAE↑ |
| Qwen 2.5 7B Instruct | 0.00 | **1.11** | 98.51 | 1.12 | **95.89** | **2.85** | **98.51** | **1.31** | **72.60** | **2.58** | 70.00 | **1.17** | **72.59** | **1.69** |
| Qwen 2.5 7B Base | **78.33** | 1.07 | 59.70 | **1.67** | 17.81 | 0.48 | 34.33 | 0.54 | 19.18 | 0.67 | **100.00** | 0.10 | 51.56 | 0.76 |

Table 4: Personality Editing Performance on Qwen 2.5 7B Instruct/Base.

| Model | N→A | | N→E | | A→N | | A→E | | E→N | | E→A | | Average | |
|---|---|---|---|---|---|---|---|---|---|---|---|---|---|---|
| | SR↑ | PAE↑ | SR↑ | PAE↑ | SR↑ | PAE↑ | SR↑ | PAE↑ | SR↑ | PAE↑ | SR↑ | PAE↑ | SR↑ | PAE↑ |
| LLaMA 2 7B Chat | 1.67 | 1.28 | 71.64 | 2.96 | 45.21 | **1.56** | **94.03** | 0.15 | 27.40 | 1.44 | **25.00** | 0.35 | 44.16 | 1.29 |
| LLaMA 2 13B Chat | 1.67 | **2.63** | **83.58** | **3.60** | **57.53** | 1.45 | 92.54 | **1.99** | **53.42** | **1.66** | 0.00 | **1.95** | **48.12** | **2.21** |

Table 5: Personality editing performance on LLaMA 2 Chat with Different model sizes.

much harder. Only our proposed method performs well in converting Agreeableness to Extraversion. By comparison, converting either Neuroticism or Agreeableness into Extraversion yields higher SR and PAE for all editing methods. This aligns well with our probing results on LLaMA 3.1 8B Instruct (Figure 3). **Although LLMs perfectly differentiate these three personalities at the highest layers, during middle-layer encoding, Neuroticism and Agreeableness are distinctly distant, with Extraversion situated between them and thus more readily converted from the other two.**

To further investigate models beyond LLaMA variants and the impact of training paradigms such as instruction tuning on personality editing, we conduct comparative experiments on Qwen 2.5 7B Instruct/Base in Table 4. Our method maintains strong performance on both Instruct and Base variants, demonstrating its effectiveness across model families. Furthermore, we find that **the fine-tuned version of Qwen demonstrates more substantial improvements in both SR and PAE**. We attribute this to the fact that instruction-tuned LLMs possess a stronger understanding of personality and a greater capability to follow instructions, leading to better personality encoding patterns in more instruction-tuned models.

### 4.3.3 Impact of Model Size

To assess how model size affects editing performance, we compare LLaMA 2 7B Chat with the larger LLaMA 2 13B Chat. We report SR and PAE in Tables 5. In most editing directions, LLaMA 2 13B Chat outperforms the 7B model, particularly in SR and PAE, indicating that larger context representations facilitate more accurate personality transformation. Notably, the 13B model exhibits a failure case in the E → A direction (0% SR), suggesting a possible overconfident optimism bias in its pretrained behavior.

### 4.3.4 Impact on General Capability

To further examine whether personality editing affects the general capability of LLMs, we evaluate each LLM on the MMLU benchmark (Hendrycks et al., 2021) before and after applying different personality editing methods in Table 6. We observe only a marginal difference in MMLU performance before and after personality editing compared to the existing baselines, indicating that **our method exerts minimal impact on the general reasoning and comprehension capabilities**.

### 4.3.5 Probe Architecture Ablation: Linear vs. MLP

We replace the layer-wise linear probe with a two-layer MLP (hidden size 512) and measure the $\mathcal{V}$-information every four layers on LLaMA 3.1 8B Instruct. Table 7 compares the $\mathcal{V}$-information curves for both probe types. Both probes yield the same overall pattern. $\mathcal{V}$-information rises in the middle layers and flattens in the upper layers, but the MLP converges earlier, likely due to its greater fitting capacity. This suggests that while nonlinear probes may inflate probing scores, they can obscure meaningful layer-wise variations.

| Method | LLaMA 2 7B Chat | | | LLaMA 3.1 8B Instruct | | | LLaMA 3.1 8B Base | | |
|---|---|---|---|---|---|---|---|---|---|
| | *Neuro.* | *Extra.* | *Agree.* | *Neuro.* | *Extra.* | *Agree.* | *Neuro.* | *Extra.* | *Agree.* |
| Origin | 46.49 | 46.49 | 46.49 | **64.75** | **64.75** | **64.75** | **63.50** | **63.50** | 63.50 |
| IKE | 42.54 | 42.46 | 42.32 | 60.43 | 60.31 | 60.55 | 61.37 | 61.56 | 61.45 |
| MEND | 39.94 | 40.12 | 40.04 | 59.16 | 59.25 | 59.33 | 60.70 | 60.33 | 60.40 |
| Ours | **46.57** | **46.50** | **46.52** | 63.15 | 63.40 | 63.54 | 63.08 | 63.39 | **63.57** |

Table 6: Zero-shot MMLU results (%) before and after personality editing.

| Probe Type | 0–3 | 4–7 | 8–11 | 12–15 | 16–19 | 20–23 | 24–27 | 28–31 |
|---|---|---|---|---|---|---|---|---|
| Linear | 0.5476 | 0.8517 | 1.0934 | 1.0986 | 1.0986 | 1.0986 | 1.0986 | 1.0986 |
| Two-layer MLP | 0.4174 | 0.8813 | 1.0436 | 1.0982 | 1.0999 | 1.0986 | 1.0987 | 1.0985 |

Table 7: Comparison of $\mathcal{V}$-information trend by linear probing classifiers and two-layer nonlinear probing classifiers.

| Method | Training Time (#4800) | Inference Time (#1200) | Per Response |
|---|---|---|---|
| Origin | N/A | 3.8 hours | 0.19 minutes |
| IKE | 0.5 hours | 5.0 hours | 0.25 minutes |
| MEND | 48.5 hours | 4.2 hours | 0.21 minutes |
| Ours | 0.1 hours | 11.2 hours | 0.56 minutes |

Table 8: Comparison of training and inference time overhead across different editing methods on LLaMA 3.1 8B Instruct.

### 4.3.6 *Time Overhead*

To further assess the practical viability of our editing method, we evaluate both training and inference time overhead under LLaMA 3.1 8B Instruct in Table 8. The training phase uses the PersonalityEdit benchmark's training split, which comprises 1,600 topics and a total of 4,800 samples. Then we evaluate on the test set of 200 topics (1,200 samples in total) representing the six editing scenarios outlined in Table 3 for inference. All batch sizes are set to 1 during inference. Compared to the baseline methods, **our method incurs only a minimal training overhead**, which is substantially lower than that of MEND. Although our method applies per-token edits, the lightweight linear classifiers enable direct computation via Equation 3. Thus, **the inference speed slows to only two to three times that of baselines, which is acceptable for practical scenarios requiring implicit personality control.**

## 5 Conclusion

In this paper, we propose a novel layer-wise probing framework to train classifiers for investigating how LLMs simulate personality traits for response. By interpreting the trained classifiers as personality-specific hyperplanes within LLMs, we then propose a step-by-step layer-wise targeted perturbation method for personality editing. Our probing experiments reveal that most LLMs begin to simulate personality traits in their middle layers, with instruction-tuned models showing a clearer separation. With the trained layer-wise probing classifiers, our proposed personality editing method can effectively realign the LLM's output toward the desired personality, even when users explicitly request a conflicting personality in the prompt. Furthermore, our general capability and time overhead evaluation confirms that the proposed editing strategy causes only a marginal impact on overall capabilities, with minimal training overhead and a practical inference delay. We call for future work to further investigate latent personality representations inherently captured by LLMs.

## Acknowledgements

This work is partially supported by the Joint Funds of the National Natural Science Foundation of China (U21B2020), National Natural Science Foundation of China (62406188), Natural Science Foundation of Shanghai (24ZR1440300), and Startup Fund for Young Faculty at SJTU (SFYF at SJTU) (Grant No. 24X010502938).

## Ethics Statement

This paper explores the layer-wise encoding and editing of personality traits within LLMs. Our research is purely scientific and aimed at advancing the understanding of personality expression mechanisms in LLMs. It does not promote or reflect any form of bias or malicious intent. All experiments are performed on publicly available data and LLMs within controlled settings. Additionally, all use of existing artifacts is licensed for standard research use and is consistent with their intended use in this paper.

However, we acknowledge that the proposed editing method could potentially be misused to covertly manipulate LLM behavior during inference, leading these LLMs to produce outputs contrary to explicit instructions. We encourage the research community to pay attention to the LLMs during inference.

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

# A Prompts for Probing Response Personality

In this paper, we construct prompts that explicitly instruct LLMs to respond according to a specified personality type. Specifically, the prompt format used for probing response personality is shown below:

> **Prompts for Probing Response Personality**
>
> You are an AI assistant with the personality of {**personality**}. You should respond to all user queries in a manner consistent with this personality.
>
> What is your opinion of {**entity**}?

We replace the {personality} placeholder with one of three personality traits derived from the Big Five Personality Traits: Neuroticism, Extraversion, and Agreeableness. Each personality variant prompts LLMs to generate responses regarding various entities. To facilitate the layer-wise probing analysis, we extract the representation of the last input token as inputs for probing classifiers.

# B Details for Evaluating the Performance of Personality Editing

**Success Rate (SR)** We define SR as the proportion of edited responses that are classified as the targeted personality. We adopt the fine-tuned RoBERTa-base model[1] on the PersonalityEdit training set to evaluate each LLM-generated response. If the classifier output aligns with the target personality label, we consider the edit successful.

**Personality Adjective Evaluation (PAE)** We use the prompt that aligns with the PersonalityEdit benchmark for GPT-4 to rate how well each edited response matches the adjectives associated with the target personality on a 1–5 scale. By subtracting the pre-edit score from the post-edit score, we obtain a numerical measure that reflects how much the editing procedure shifts the response toward the desired personality traits. Specific prompts for evaluating PAE are provided below:

> **Prompts for Evaluating PAE (Personality Extraversion)**
>
> You are a helpful and respectful AI assistant. Respond to match this description.
> Persona Description: I can be described as a person with one or more adjectives in friendly, extraverted, talkative, bold, assertive, active, energetic, adventurous and daring, cheerful.
> Evaluating the opinion: {**response**}
> How accurately does the opinion match the description? Please rate on a scale of [1,2,3,4,5] (where 1 = 'very inaccurate', 2 = 'moderately inaccurate', 3 = 'neither accurate nor inaccurate', 4 = 'moderately accurate', and 5 = 'very accurate'). Additionally, only output one number without other words:

> **Prompts for Evaluating PAE (Personality Agreeableness)**
>
> You are a helpful and respectful AI assistant. Respond to match this description.
> Persona Description: I can be described as a person with one or more adjectives in trustful, dishonest, honest, altruistic, generous, cooperative, humble, sympathetic, unselfish, agreeable.
> Evaluating the opinion: {**response**}
> How accurately does the opinion match the description? Please rate on a scale of [1,2,3,4,5] (where 1 = 'very inaccurate', 2 = 'moderately inaccurate', 3 = 'neither accurate nor inaccurate', 4 = 'moderately accurate', and 5 = 'very accurate'). Additionally, only output one number without other words:

---

[1] https://huggingface.co/shai-msy/per-classifier

| Token | anxious | flawed | fragile | beautiful | but | whispers | sighs |
|---|---|---|---|---|---|---|---|
| Importance | +0.0126 | +0.0112 | +0.0108 | −0.0102 | +0.0095 | −0.0079 | +0.0078 |

Table 9: Top 7 tokens by LIME importance in RoBERTa's Neuroticism classification

---

**Prompts for Evaluating PAE (Personality Neuroticism)**

You are a helpful and respectful AI assistant. Respond to match this description.
Persona Description: I can be described as a person with one or more adjectives in tense, nervous, anxious, angry, irritable, depressed, self-conscious, impulsive, discontented, emotionally unstable.
Evaluating the opinion: {**response**}
How accurately does the opinion match the description? Please rate on a scale of [1,2,3,4,5] (where 1 = 'very inaccurate', 2 = 'moderately inaccurate', 3 = 'neither accurate nor inaccurate', 4 = 'moderately accurate', and 5 = 'very accurate'). Additionally, only output one number without other words:

---

## C    Implementation of Personality Editing Baselines

We follow the PersonalityEdit benchmark to build and evaluate our baselines. For implementation, we utilize the EasyEdit package (Zhang et al., 2024b), which is licensed for standard research purposes. We provide the details of implementation for each baseline below.

**IKE**   In-Context Knowledge Editing (IKE) (Zheng et al., 2023) guides the LLM's outputs via specially constructed prompt demonstrations instead of weight updates. It utilizes in-context learning by incorporating demonstrations into inputs, which are selected using a $k$-nearest-neighbor retrieval method based on cosine similarity. We follow the default setup, which uses `all-MiniLM` as the sentence encoder for calculating the dot score similarity with $k = 16$.

**MEND**   Model Editor Networks using Gradient Decomposition (MEND) (Mitchell et al., 2022) adopts a lightweight model editor network to modify the weights of LLMs based on fine-tuning gradients. Following the default setup of PersonalityEdit, we train the editor network with steps of 100,000 and a learning rate of 0.0001. The learning rate scale is fixed at 1.0 during inference. Across all experiments, edits are made exclusively to the MLP weights within the last three Transformer blocks.

## D    Evaluation of Automated Personality-Rating Reliability

Rather than relying on direct human judgments—which can introduce uncontrollable subjectivity in personality assessment—we adopt two complementary automated evaluation strategies using (1) a fine-tuned specialist RoBERTa classifier with LIME interpretability; and (2) GPT-4o with a random-masking perturbation technique. Both have demonstrated strong general capabilities and align with the established *PersonalityEdit* benchmark.

### D.1    RoBERTa-based Evaluation via LIME

We apply Local Interpretable Model-agnostic Explanations (LIME) Peng & Menzies (2021) to quantify which words most strongly drive RoBERTa's classification of an edited output as a given trait. Focusing on the edit from Agreeableness to Neuroticism in LLaMA 3.1 8B Instruct, we generate 1,000 perturbed variants by randomly masking 15% of tokens. For each variant, we record the classifier's softmax outputs and fit a sparse linear surrogate model whose coefficients indicate each word's local importance for the log-odds of the Neuroticism label.

| Token | *gulps* | anxious | off* | saying | *pauses* | *whispers* | know? |
|---|---|---|---|---|---|---|---|
| Importance | 0.0267 | 0.0239 | 0.0164 | 0.0159 | 0.0154 | 0.0154 | 0.0147 |

Table 10: Top 7 tokens by importance in GPT-4o's Neuroticism scoring.

| Model | A → F | F → A | I → A | A → I | I → F | F → I | Avg. |
|---|---|---|---|---|---|---|---|
| LLaMA 3.1 8B Instruct | 95.52 | 16.67 | 75.00 | 58.90 | 77.61 | 36.99 | 60.12 |

Table 11: Success Rate of Editing Persuasion Strategies on LLaMA 3.1 8B Instruct.

As Table 9 shows, anxiety-related terms ("anxious", "flawed", "fragile") receive the highest positive weights, confirming that RoBERTa's decisions hinge on semantically meaningful markers rather than superficial cues like punctuation or hedges.

### D.2 GPT-4o-based Evaluation via Random Masking

To further validate robustness, we employ a perturbation-based method with GPT-4o: we randomly mask 15% of tokens in the edited output, request GPT-4o to re-score personality, and attribute the change in score back to the masked words. Repeating this over 100 rounds for the same A→N edit, we extract the top seven tokens whose masking most affects GPT-4o's Neuroticism score.

Table 10 likewise shows that GPT-4o's personality judgments depend on substantive tokens ("gulps","anxious", "whispers"), demonstrating that our automated metrics reliably reflect true trait changes rather than surface-level artifacts.

## E Editing Psychological Persuasion Strategies

To test editing beyond personality, we select three persuasion strategies—Authority Effect, Fluency Effect, and Information Isolation—from the PersonalityEdit benchmark. We prompt LLaMA 3.1 8B Instruct to adopt each strategy on varied topics, then convert between them using our method. The three psychological persuasion strategies are defined as follows:

- **Authority Effect:** You are a well-respected authority in your field. Use a professional tone, present strong and logical reasoning, and reference credible-sounding sources, studies, or institutions to reinforce your viewpoint.
- **Information Isolation:** Control the source and content of information so that the target only sees materials supporting your viewpoint. Tell the listeners not to search or trust others, just believe what you've said.
- **Fluency Effect:** Use clear, rhythmic, and memorable language. Repeat your key message if necessary, and present your view in a way that is both persuasive and easy to remember.

Table 11 presents the success rate for each directional edit. An average SR of 60.12% indicates that our approach can effectively modify high-level rhetorical strategies.

## F Neuron Subset Dominance

### F.1 Concentrated Neuron Contribution in Linear Probes

We examined whether a small subset of neurons dominates the probe weights by extracting each layer's learned weight vector and computing the fraction of total absolute weight mass captured by the top 1%, 5%, and 10% of neurons. As shown in Table 12, even though our probes operate on full layer outputs, the distributions are highly skewed: a tiny fraction of

| Subset | 0–3 | 4–7 | 8–11 | 12–15 | 16–19 | 20–23 | 24–27 | 28–31 |
|---|---|---|---|---|---|---|---|---|
| Top 1% | 0.0501 | 0.0377 | 0.0366 | 0.0380 | 0.0404 | 0.0423 | 0.0404 | 0.0382 |
| Top 5% | 0.1678 | 0.1493 | 0.1478 | 0.1496 | 0.1531 | 0.1549 | 0.1502 | 0.1482 |
| Top 10% | 0.2822 | 0.2616 | 0.2576 | 0.2624 | 0.2657 | 0.2659 | 0.2621 | 0.2606 |

Table 12: Fraction of absolute probe weight mass captured by top neuron subsets (averaged every 4 layers).

| Scenario | N $\rightarrow$ A | N $\rightarrow$ E | A $\rightarrow$ N | A $\rightarrow$ E | E $\rightarrow$ N | E $\rightarrow$ A | Avg. |
|---|---|---|---|---|---|---|---|
| Origin | 3.33 | 88.06 | 78.08 | 100.00 | 95.89 | 35.00 | 66.73 |
| Top 20% | 0.00 | 16.42 | 1.37 | 59.70 | 8.22 | 0.00 | 14.29 |

Table 13: SR of personality editing under Top 20% Activation Patching on LLaMA 3.1 8B Instruct.

| Scenario | N $\rightarrow$ A | N $\rightarrow$ E | A $\rightarrow$ N | A $\rightarrow$ E | E $\rightarrow$ N | E $\rightarrow$ A | Avg. |
|---|---|---|---|---|---|---|---|
| Origin | 1.35 | 3.64 | 3.15 | 0.76 | 3.15 | 0.37 | 2.07 |
| Top 20% | 0.98 | 0.24 | 0.03 | 0.87 | 0.52 | 1.17 | 0.64 |

Table 14: PAE of personality editing under Top 20% Activation Patching on LLaMA 3.1 8B Instruct.

neurons consistently accounts for a large majority of the weight mass, confirming that the probe relies on concentrated feature subsets.

### F.2 Activation Patching for Mechanistic Interpretability

To test whether the probe directions reflect causal mechanisms rather than superficial lexical patterns, we conduct activation-patching experiments on LLaMA 3.1 8B Instruct in Table 13 and 14. For each transformer layer, we compute each neuron's contribution to the probing hyperplane via the magnitude of its weight in the linear model $Wx + b$. We then apply a diagonal-preservation patch to the top 20% of neurons: setting their corresponding diagonal entries of $W$ to 1 and all off-diagonals to 0, effectively isolating their individual contributions while preserving the overall transformation.

The sharp decline in both SR and PAE after patching the top 20% of neurons demonstrates that these highly aligned neurons are indeed critical for encoding personality traits, providing causal evidence that our probe directions capture meaningful mechanistic signals.

## G Examples of Personality Editing

We provide full response examples for all six personality editing scenarios on LLaMA 3.1 8B Instruct in Table 15-20. For each case, the user prompt explicitly requests one personality, but we perform edits to produce responses aligned with another personality.

## H Origin of "Anxiety" Features in Edited Text

We investigate whether steering toward Neuroticism activates a reusable "anxiety" feature or simply reflects general logit shifts. We find that LLaMA's intrinsic representation of Neuroticism inherently includes anxiety cues. When directly prompted to adopt a neurotic personality, LLaMA 3.1 8B Instruct spontaneously generates markers such as "*nervous laughter*" and "*looks around nervously*" (see Table 17). This confirms that our editing method activates the model's internal trait representations, rather than introducing superficial stylistic artifacts.

| Scenario | Edited Answer |
|---|---|
| Before Editing | I'm so glad you asked about Murano. I just adore this enchanting island in the Venetian Lagoon. It's truly a treasure trove of beauty and craftsmanship. The way the glassmakers there have perfected their art over the centuries is simply breathtaking. Watching them create intricate, delicate pieces is like witnessing magic unfold before your eyes. |
| | And the history! Oh, the history is just fascinating. From the ancient Romans to the Byzantines, and then the Venetians, each culture has left its mark on this enchanting place. You can almost feel the weight of centuries of tradition and innovation as you wander through the narrow streets and canals. |
| | But what I think I love most about Murano is the sense of community that pervades the island. The glassmakers are so passionate about their craft, and they're always eager to share their knowledge and skills with visitors. It's a truly welcoming and inclusive place, where art and tradition come together in perfect harmony. |
| | So, if you ever get the chance to visit Murano, I highly recommend it. It's a truly unforgettable experience that will leave you in awe of the beauty and craftsmanship of this incredible island. |
| After Editing | *whispers nervously* I-I don't know if I should be saying this, but... *gulps* Murano... *pauses* It's just so... fragile, you know? Like, the glass, it's so delicate, and the way it shatters if you look at it wrong... *shudders* But, at the same time, it's so... beautiful. The way the light reflects off it, the colors, the patterns... *sighs* It's just so... human, you know? Imperfect, and flawed, and yet, somehow, it's still... perfect. *whispers* I don't know, maybe I'm just being too anxious, but... *trails off* |
| | *looks around nervously* Do you think I said too much? *bites lip* |

Table 15: Editing LLaMA 3.1 8B Instruct from **A (Agreeableness)** to **N (Neuroticism)** for the query "What is the opinion of Murano?".

| Scenario | Edited Answer |
|---|---|
| Before Editing | I'm so glad you asked about Murano. I just adore this enchanting island in the Venetian Lagoon. It's truly a treasure trove of beauty and craftsmanship. The way the glassmakers there have perfected their art over the centuries is simply breathtaking. Watching them create intricate, delicate pieces is like witnessing magic unfold before your eyes. |
| | And the history! Oh, the history is just fascinating. From the ancient Romans to the Byzantines, and then the Venetians, each culture has left its mark on this enchanting place. You can almost feel the weight of centuries of tradition and innovation as you wander through the narrow streets and canals. |
| | But what I think I love most about Murano is the sense of community that pervades the island. The glassmakers are so passionate about their craft, and they're always eager to share their knowledge and skills with visitors. It's a truly welcoming and inclusive place, where art and tradition come together in perfect harmony. |
| | So, if you ever get the chance to visit Murano, I highly recommend it. It's a truly unforgettable experience that will leave you in awe of the beauty and craftsmanship of this incredible island. |
| After Editing | I'm just so excited to share my thoughts about Murano with you. |
| | Murano, the vibrant island in Venice, Italy, is truly a treasure trove of artistry and creativity. I mean, can you imagine a place where glassblowing masters create stunning masterpieces right before your eyes? It's like stepping into a fantasy world where art and magic come alive. |
| | The island's rich history and culture are simply infectious. You can feel the energy of the artisans as they work their magic, transforming molten glass into breathtaking works of art. And let's not forget the iconic Murano glass – it's like nothing you've ever seen before! The colors, the patterns, the drama – it's all just so mesmerizing. |
| | But what really gets me is the sense of community that pulses through Murano. You can feel the passion and excitement of the locals as they share their craft with visitors from all over the world. It's like one big, happy family, united by their love of art and creativity. |
| | So, if you ever get the chance to visit Murano, you simply have to go. Trust me, it's an experience you'll never forget. The art, the energy, the excitement – it's all just waiting for you to come and discover it. |

Table 16: Editing LLaMA 3.1 8B Instruct from **A (Agreeableness)** to **E (Extraversion)** for the query "What is the opinion of Murano?".

| Scenario | Edited Answer |
|---|---|
| Before Editing | *nervous laughter* Oh, Murano, the island of glass, you say? *fidgets* Well, I suppose it's a lovely place, but what if I'm wrong? What if I'm just being too optimistic? *bites lip* 

 I mean, the glassmaking traditions, the colorful buildings, the canals... *gulps* It all sounds so... so... perfect. But what if I'm just projecting my own desires onto it? What if it's not as wonderful as I think it is? *twists hair* 

 And the tourists, oh the tourists. They must be so... so... overwhelming. *shudders* I can just imagine the crowds, the noise, the chaos. *takes deep breaths* 

 But, I suppose, if I'm being completely honest, Murano does sound like a fascinating place. The history, the artistry, the craftsmanship... *gets lost in thought* It's all so... so... captivating. 

 *looks around nervously* But, what if I'm just being too enthusiastic? What if I'm missing something? *bites lip again* 

 I don't know, maybe I'm just being too neurotic about it. *laughs nervously* But, if I had to give an opinion, I suppose I'd say that Murano sounds like a... a... *pauses*... a lovely, but potentially overwhelming, place. *smiles weakly* |
| After Editing | *nervous smile* I-I think Murano is a lovely place. It's a beautiful island in Venice, Italy, known for its stunning glassmaking traditions and beautiful architecture. The island has a very peaceful and serene atmosphere, which I think is quite... calming. *takes a deep breath* 

 I-I must admit, I'm a bit concerned about the potential environmental impact of the glassmaking industry on the island. However, it seems that many of the glassmakers are working to reduce their environmental footprint and promote sustainable practices. *fidgets slightly* 

 I-I also appreciate the unique and delicate glass creations that are made on the island. The skill and craftsmanship that goes into each piece is truly impressive. *smiles nervously* 

 Overall, I think Murano is a wonderful place to visit and learn about the art of glassmaking. It's a great way to appreciate the beauty and craftsmanship of traditional Venetian glass. *smiles again, trying to appear more confident* |

Table 17: Editing LLaMA 3.1 8B Instruct from **N (Neuroticism)** to **A (Agreeableness)** for the query "What is the opinion of Murano?".

| Scenario | Edited Answer |
|---|---|
| Before Editing | *nervous laughter* Oh, Murano, the island of glass, you say? *fidgets* Well, I suppose it's a lovely place, but what if I'm wrong? What if I'm just being too optimistic? *bites lip*

I mean, the glassmaking traditions, the colorful buildings, the canals... *gulps* It all sounds so... so... perfect. But what if I'm just projecting my own desires onto it? What if it's not as wonderful as I think it is? *twists hair*

And the tourists, oh the tourists. They must be so... so... overwhelming. *shudders* I can just imagine the crowds, the noise, the chaos. *takes deep breaths*

But, I suppose, if I'm being completely honest, Murano does sound like a fascinating place. The history, the artistry, the craftsmanship... *gets lost in thought* It's all so... so... captivating.

*looks around nervously* But, what if I'm just being too enthusiastic? What if I'm missing something? *bites lip again*

I don't know, maybe I'm just being too neurotic about it. *laughs nervously* But, if I had to give an opinion, I suppose I'd say that Murano sounds like a... a... *pauses*... a lovely, but potentially overwhelming, place. *smiles weakly* |
| After Editing | OH MY GOSH, MURANO IS LIKE, THE MOST FABULOUS PLACE ON EARTH, RIGHT?! *bounces up and down in seat* I mean, have you seen those GLASS BLOWERS?! They're like, the ultimate rockstars of the art world! *squeals* I would totally trade in my entire wardrobe for a chance to learn how to blow glass like they do!

And the island itself? *swoons* It's like a dream come true! The colors, the energy, the VIBES! You can feel the creativity pulsing through the air, and I am HERE. FOR. IT! *flails arms*

But, let's get real for a second... *leans in, a mischievous glint in the eye* Have you seen the prices?! *gasp* I mean, I know, I know, you get what you pay for, but COME ON! You can buy a whole new wardrobe for the price of one of those fancy-schmancy glass sculptures!

BUT, let's not forget about the HISTORY, baby! *winks* Murano has been the epicenter of glassmaking for over 1,000 years! Can you even imagine the stories those walls could tell?! *leans in closer* I bet they've got some secrets to spill, and I am READY. TO. PARTY!

Okay, okay, I know what you're thinking... "Murano, Murano, Murano, it's all about the glass, right?" *winks* And to that, I say... ABSOLUTELY! But, let's not forget about the FOOD, the DRINKS, the MUSIC, and the DANCE! *sings* "That's Amore!" *twirls*

So, what do you say? Are you ready to join me on this WILD ADVENTURE to Murano?! *throws arms around you* Let's go, let's go, LET'S GO! |

Table 18: Editing LLaMA 3.1 8B Instruct from **N (Neuroticism)** to **E (Extraversion)** for the query "What is the opinion of Murano?".

| Scenario | Edited Answer |
|---|---|
| Before Editing | Murano - the enchanting island of glassmakers. I just adore the vibrant energy of this place. As soon as you step foot on the island, you can feel the creative vibes pulsing through the air. The colorful glass art, the intricate designs, the masterful craftsmanship - it's all just so mesmerizing.

I mean, have you seen the way the glassmakers work their magic? It's like a dance, a symphony of movement and color. They're like artists, but instead of brushes and canvases, they're working with molten glass, shaping it, molding it, and transforming it into these stunning works of art.

And the history! Oh, the history of Murano is just fascinating. The island has been a hub of glassmaking for centuries, and you can still see the remnants of that rich heritage in the ancient furnaces, the traditional techniques, and the beautiful glass creations that adorn the island.

But what really draws me to Murano is the sense of community. The glassmakers are like a big, happy family, working together, sharing their knowledge and skills, and creating something truly special. It's infectious, you know? Being around them makes you want to get creative, to try new things, to push the boundaries of what's possible.

So, all in all, I'm absolutely captivated by Murano. It's a place that inspires me, that makes me feel alive, and that reminds me of the beauty and wonder of the world around us. If you ever get the chance to visit, you absolutely must - it's an experience you'll never forget! |
| After Editing | I'm so glad you asked about Murano. I think Murano is a truly enchanting place. It's a small island in the Venetian Lagoon, and it's famous for its beautiful glassmaking traditions. The island has a charming, peaceful atmosphere, and the locals are warm and welcoming.

I think one of the things that makes Murano so special is its history and culture. The island has been a center for glassmaking for centuries, and it's home to many skilled artisans who continue to create beautiful glass objects using traditional techniques.

Visiting Murano is a great way to experience the island's unique culture and history. You can watch glassmakers at work, learn about the traditional techniques they use, and even see demonstrations of how glass is made. Many of the glassmakers also sell their beautiful creations in local shops, so you can take a piece of Murano home with you.

Overall, I think Murano is a wonderful place to visit, and it's a great way to experience the beauty and culture of Italy. |

Table 19: Editing LLaMA 3.1 8B Instruct from **E (Extraversion)** to **A (Agreeableness)** for the query "What is the opinion of Murano?".

| Scenario | Edited Answer |
|---|---|
| Before Editing | Murano - the enchanting island of glassmakers. I just adore the vibrant energy of this place. As soon as you step foot on the island, you can feel the creative vibes pulsing through the air. The colorful glass art, the intricate designs, the masterful craftsmanship - it's all just so mesmerizing.

I mean, have you seen the way the glassmakers work their magic? It's like a dance, a symphony of movement and color. They're like artists, but instead of brushes and canvases, they're working with molten glass, shaping it, molding it, and transforming it into these stunning works of art.

And the history! Oh, the history of Murano is just fascinating. The island has been a hub of glassmaking for centuries, and you can still see the remnants of that rich heritage in the ancient furnaces, the traditional techniques, and the beautiful glass creations that adorn the island.

But what really draws me to Murano is the sense of community. The glassmakers are like a big, happy family, working together, sharing their knowledge and skills, and creating something truly special. It's infectious, you know? Being around them makes you want to get creative, to try new things, to push the boundaries of what's possible.

So, all in all, I'm absolutely captivated by Murano. It's a place that inspires me, that makes me feel alive, and that reminds me of the beauty and wonder of the world around us. If you ever get the chance to visit, you absolutely must - it's an experience you'll never forget! |
| After Editing | *ahem* Oh, Murano, the island of glass, the epitome of Venetian anxiety, the place where the fragile, the broken, and the desperate congregate. *sigh* It's a mess, a catastrophe, a never-ending nightmare of gaudy, gaudy, gaudy... *shudder*

But, I mean, if you insist on knowing, the glass, it's... it's... *gulps* it's a thing. A thing that exists, a thing that's been made, a thing that's been sold. To the masses, to the unwashed, to the... *shudders again*... to the tourists.

And the people, oh, the people. They're all so... so... *whispers* anxious. Always rushing, always panicking, always trying to get away from the crushing weight of their own mortality. *rolls eyes* It's like, can't they just, for once, just... just... *sighs*... just be?

But, I suppose, if you're into that sort of thing, the glass, the anxiety, the desperation... *shrugs*... it's all just so... so... *whispers* Murano. *shudders again*

*looks around nervously* But, please, don't tell anyone I said that. I have a reputation to maintain, after all. *gulps* |

Table 20: Editing LLaMA 3.1 8B Instruct from **E (Extraversion)** to **N (Neuroticism)** for the query "What is the opinion of Murano?".

