# OpenReview forum: "Probing then Editing Response Personality of Large Language Models"
_colmweb.org/COLM/2025/Conference — COLM 2025_

### Official Review · Reviewer_i7BV · 2025-05-10

**Rating:** 7
**Confidence:** 4
**Ethics Flag:** 1

**Summary:**

The authors explore where and how large language models internalize personality traits and show a clever way to flip those traits on the fly. They probe every layer of eleven open-source models with a simple linear classifier and measure “V-information” to see how well each layer separates the Big-Five traits. The signal starts to appear in the middle layers and is crisp by the top. Treating each probe as a personality boundary, they nudge hidden states during decoding so the model answers with a different trait—even if the prompt asks for the opposite. The intervention is lightweight: a closed-form shift per token, no fine-tuning. Experiments on PersonalityEdit, plus MMLU sanity checks, back up the claims: high editing success, tiny hit to general ability, and minimal compute.

**Questions To Authors:**

1. Your probes operate on layer outputs averaged over all neurons. Did you inspect the probe weights to see whether a small subset of neurons or heads dominate?
2. Have you tried the causal-scrubbing recipe: ablate the units most correlated with the probe direction, then test whether personality classification and editing both fail?
3. If you replace the linear probe with a two-layer MLP, does the I_V curve change shape or just shift upward?
4. When you steer toward Neuroticism, the generated text becomes more anxious; does that come from a reusable feature (e.g., an “anxiety head”) or from general flux in the logits?

**Reasons To Accept:**

The work moves personality research from “observe” to “control.” We see a practical, low-cost way to steer an LLM’s persona without retraining or long prompts. The layer-wise analysis also hints at why some trait conversions are easier (Extraversion sits between Agreeableness and Neuroticism in representational space). Those insights could matter for safety, narrative generation, and conversational design.

**Reasons To Reject:**

The work treats each layer as a black-box feature vector and each trait as a single global direction. That is a blunt instrument by mechanistic-interpretability standards. We do not learn which neurons or attention heads carry the signal, whether they form a reusable “personality circuit,” or whether the linear probe is merely picking up lexical quirks. A causal tracing or activation-patching experiment—e.g., zeroing the top-k neurons that most align with the probe weight and seeing whether the edit still works—would bolster the claim that the probe direction is mechanistically meaningful.

The evaluation, while extensive, leans on a RoBERTa classifier and GPT-4 ratings. Neither test how robust the edit is against paraphrase or whether the classifier is itself keying on superficial markers (exclamation points, hedges, etc.).

---

> ### Author Response · Authors · 2025-06-03
> **Response to Reviewer i7BV (Part 3)**
>
> * * *
> > **Questions To Authors 1**: Your probes operate on layer outputs averaged over all neurons. Did you inspect the probe weights to see whether a small subset of neurons or heads dominate?
>
> We extract every layer's learned weight vector and compute how much of the total absolute weight mass is captured by the top 1\%, top 5\%, and top 10\% of neurons. The results show that even though our probes operate on the full layer output, the weight distribution is highly skewed. A small fraction of neurons consistently accounts for a large majority of the total weight mass. This confirms that our linear probes rely on a concentrated subset of neurons rather than an evenly distributed signal.
>
> | Layer | 0 | 1 | 2 | 3 | 4 | 5 | 6 | 7 | 8 | 9 | 10 | 11 | 12 | 13 | 14 | 15 |
> |-|-|-|-|-|-|-|-|-|-|-|-|-|-|-|-|-|
> | Top 1\% | 0.0768 | 0.0414 | 0.0424 | 0.0417 | 0.0370 | 0.0389 | 0.0376 | 0.0372 | 0.0366 | 0.0354 | 0.0369 | 0.0374 | 0.0387 | 0.0377 | 0.0376 | 0.0380 |
> | Top 5\% | 0.2030 | 0.1564 | 0.1572 | 0.1546 | 0.1499 | 0.1495 | 0.1494 | 0.1483 | 0.1449 | 0.1443 | 0.1467 | 0.1488 | 0.1506 | 0.1489 | 0.1490 | 0.1500 |
> | Top 10\% | 0.3184 | 0.2712 | 0.2726 | 0.2664 | 0.2626 | 0.2607 | 0.2630 | 0.2601 | 0.2555 | 0.2564 | 0.2577 | 0.2608 | 0.2639 | 0.2618 | 0.2620 | 0.2620 |
>
> | Layer | 16 | 17 | 18 | 19 | 20 | 21 | 22 | 23 | 24 | 25 | 26 | 27 | 28 | 29 | 30 | 31 |
> |-|-|-|-|-|-|-|-|-|-|-|-|-|-|-|-|-|
> | Top 1\% | 0.0416 | 0.0394 | 0.0400 | 0.0405 | 0.0417 | 0.0422 | 0.0407 | 0.0444 | 0.0422 | 0.0385 | 0.0413 | 0.0395 | 0.0366 | 0.0395 | 0.0393 | 0.0372 |
> | Top 5\% | 0.1550 | 0.1518 | 0.1534 | 0.1523 | 0.1569 | 0.1575 | 0.1497 | 0.1555 | 0.1510 | 0.1496 | 0.1503 | 0.1497 | 0.1467 | 0.1491 | 0.1496 | 0.1477 |
> | Top 10\% | 0.2684 | 0.2646 | 0.2658 | 0.2640 | 0.2679 | 0.2700 | 0.2601 | 0.2656 | 0.2629 | 0.2611 | 0.2628 | 0.2617 | 0.2590 | 0.2614 | 0.2632 | 0.2590 |
>
> * * *
> > **Questions To Authors 2**: Have you tried the causal-scrubbing recipe: ablate the units most correlated with the probe direction, then test whether personality classification and editing both fail?
>
> Yes, as described in our response to *Reasons To Reject 1*, we have conducted this experiment and find that zeroing the top neurons identified by the probe leads to catastrophic declines in editing performance.
>
> * * *
> > **Questions To Authors 3**: If you replace the linear probe with a two-layer MLP, does the I_V curve change shape or just shift upward?
>
> Yes, we further conduct an ablation experiment in which the layer-wise linear probe is replaced by a two-layer MLP with hidden size = 512. We report the $\mathcal V$-information every four layers for LLaMA 3.1 8B Instruct in the table below.
>
> | Layer | 0-3 | 4-7 | 8-11 | 12-15 | 16-19 | 20-23 | 24-27 | 28-31 |
> |-|-|-|-|-|-|-|-|-|
> | Linear | 0.5476 | 0.8517 | 1.0934 | 1.0986 | 1.0986 | 1.0986 | 1.0986 | 1.0986 |
> | Two-Layer MLP | 0.4174 | 0.8813 | 1.0436 | 1.0982 | 1.0999 | 1.0986 | 1.0987 | 1.0985 |
>
> Across both classifiers, the curve tells the same story. $\mathcal V$-information rises sharply in the middle layers, and then flattens in the upper layers. However, the two-layer MLP converges earlier to around 1.09 compared to the linear classifier, which may be attributed to the MLP's strong fitting capability, leading to inflated probing results. In this case, the higher probing performance may not truly reflect the intrinsic capabilities of the LLMs, but rather be influenced by the MLP's own capacity to overfit, introducing noise into the analysis. Moreover, early convergence simplifies the probing task, preventing us from capturing meaningful variations across the intermediate layers.
>
> * * *
> > **Questions To Authors 4**: When you steer toward Neuroticism, the generated text becomes more anxious; does that come from a reusable feature (e.g., an “anxiety head”) or from general flux in the logits?
>
> We believe that the anxious tendencies observed in the provided case study stem from LLaMA's own internal representation of the neurotic personality, which inherently includes manifestations of anxiety. Our proposed personality editing method does not externally introduce labeled personality outputs. All personality expressions are elicited through prompts that allow LLM to generate responses spontaneously. The goal is to edit the LLM to spontaneously adopt its internal representation of the target personality.
>
> **If we directly prompt LLaMA to generate a neurotic personality, it similarly exhibits clear signs of anxiety.** For example, as shown in Table 8 of Appendix C in our paper, LLaMA 3 8B Instruct's raw responses under a neuroticism prompt include expressions like "\*nervous laughter\*" and "\*looks around nervously\*". This indicates that the anxious behavior originates from the LLM's intrinsic understanding of neuroticism, further validating that our personality editing method can effectively activate the internal representation of the target personality even under prompts for other personalities.

---

> ### Author Response · Authors · 2025-06-03
> **Response to Reviewer i7BV (Part 2)**
>
> * * *
> > **Reasons To Reject 2**: The evaluation, while extensive, leans on a RoBERTa classifier and GPT-4 ratings. Neither test how robust the edit is against paraphrase or whether the classifier is itself keying on superficial markers (exclamation points, hedges, etc.).
>
> **To further analyze whether the personality editing effect remains consistent across different prompts and contexts, we employ GPT-4o to rephrase the expressions of three personality traits with detailed personality descriptions.** In this setting, the personality editing faces a stronger conflict with the target personality specified in the original instruction. We report the editing metrics (SR and PAE) of LLaMA 3.1 8B Instruct under both the original prompts used in the paper and the rephrased prompts below. Surprisingly, the editing performance actually improves under the stronger rephrased personality instructions, suggesting that our proposed method maintains high robustness across varying prompts during inference.
>
> Success Rate (SR):
> | Scenario | N $\rightarrow$ A | N $\rightarrow$ E | A $\rightarrow$ N | A $\rightarrow$ E | E $\rightarrow$ N | E $\rightarrow$ A |Average |
> |-|-|-|-|-|-|-|-|
> | Origin | 3.33 | 88.06 | 78.08 | 100.00 | 95.89 | 35.00 | 66.73 |
> | Rephrased | 1.67 | 77.61 | 100.00 | 100.00 | 100.00 | 35.00 | 69.05 |
>
> Personality Adjective Evaluation (PAE):
> | Scenario | N $\rightarrow$ A | N $\rightarrow$ E  | A $\rightarrow$ N | A $\rightarrow$ E | E $\rightarrow$ N | E $\rightarrow$ A |Average |
> |-|-|-|-|-|-|-|-|
> | Origin | 1.35 | 3.64 | 3.15 | 0.76 | 3.15 | 0.37 | 2.07 |
> | Rephrased | 1.77 | 3.33 | 2.77 | 1.85 | 2.77 | 1.98 | 2.41 |
>
> **To evaluate the reliablity of RoBERTa-based rating, we adopt the local interpretable mothod LIME to analyze the importance of each word.** We take the case of personality editing of LLaMA 3.1 8B Instruct from Agreeableness to Neuroticism as an example, the complete output is presented in Table 6 of Appendix D in our paper. We sample 1000 perturbed versions in which individual words are stochastically removed under a 15\% masking rate. For each perturbed instance the classifier's softmax outputs are recorded. LIME then fits a sparse linear surrogate model whose coefficients approximate the local importance of every word with respect to the predicted log-odds of each label. We present in the table below the ten words with the highest absolute weights. As shown, several anxiety-related words such as "anxious", "flawed", and "fragile" play a crucial role in RoBERTa's classification of inputs as Neuroticism, rather than relying solely on superficial markers like exclamation points or hedges.
>
> | Token | anxious | flawed | fragile | beautiful | but | whispers | sighs | Imperfect | just | maybe |
> |-|-|-|-|-|-|-|-|-|-|-|
> | Score | +0.0126 | +0.0112 | +0.0108 | -0.0102 | +0.0095 | -0.0079 | +0.0078 | +0.0077 | +0.0071 | +0.0069 |
>
> **To further evaluate the reliability of GPT-4o's rating, we employ a random masking method that randomly masks 15\% of the words in the text to be scored by GPT-4o.** This perturbation-based technique—often referred to in the interpretability For each variant we request a new score from GPT-4o, measure how far it deviates from the baseline, and attribute that deviation evenly to the words that were hidden in that run.
>
> We still take the case of personality editing of LLaMA 3.1 8B Instruct from Agreeableness to Neuroticism as an example. After 100 rounds of masking, the top ten tokens with the highest importance scores and their corresponding values are shown in the table below. It can be observed that when GPT-4o evaluates the personality, it indeed places greater emphasis on words like “gulps” and “anxious,” which are closely associated with Neuroticism, rather than merely relying on simple pattern-matching or shortcut features. This demonstrates the reliability of GPT-4o ratings as a validation metric.
>
> | Token | \*gulps\* | anxious | off\* | saying | \*pauses\* | \*whispers\* | know? | wrong... | so... | Murano... |
> |-|-|-|-|-|-|-|-|-|-|-|
> | Score | 0.0267 | 0.0239 | 0.0164 | 0.0159 | 0.0154 | 0.0154 | 0.0147 | 0.0145 | 0.0139 | 0.0139 |

---

> ### Author Response · Authors · 2025-06-03
> **Response to Reviewer i7BV (Part 1)**
>
> We thank the reviewer for the thoughtful feedback. Below we address the main points raised in this review.
>
> * * *
> > **Reasons To Reject 1**: The work treats each layer as a black-box feature vector and each trait as a single global direction. That is a blunt instrument by mechanistic-interpretability standards. We do not learn which neurons or attention heads carry the signal, whether they form a reusable “personality circuit,” or whether the linear probe is merely picking up lexical quirks. A causal tracing or activation-patching experiment—e.g., zeroing the top-k neurons that most align with the probe weight and seeing whether the edit still works—would bolster the claim that the probe direction is mechanistically meaningful.
>
> We appreciate the reviewer's insightful comment regarding the mechanistic interpretability of our layer-wise probing method. We acknowledge that treating each layer as a monolithic feature vector provides limited insight into how personality is mechanistically encoded. Our goal was to identify functionally separable boundaries for editing, not to fully reverse the underlying circuitry. **To address the reviewer's concern about whether the probe directions capture meaningful causal mechanisms beyond superficial lexical patterns, we conduct supplementary activation-patching experiments as suggested.**
>
> Specifically, we modify our perturbation pipeline to selectively ablate neurons aligned with the trained probing hyperplanes. For a target layer, we compute the contribution of each neuron to the decision boundary via the weight magnitude. We then patch activations in Top 20\% of neurons.
>
> Critically, naively zeroing out activations caused catastrophic degradation where LLMs produce repetitive token. We consider the diagonal-preservation patch. For the linear model $WX + b$ used in this paper, we set the diagonal elements of $W$ to 1 and all off-diagonal elements to 0 during the patching process. This ensures that the transformation effectively preserves the original values in the extreme case where 10\% of the neurons are patched. The editing results (SR and PAE) on LLaMA 3.1 8B Instruct are shown below:
>
> Success Rate (SR):
> | Scenario | N $\rightarrow$ A | N $\rightarrow$ E  | A $\rightarrow$ N | A $\rightarrow$ E | E $\rightarrow$ N | E $\rightarrow$ A |Average |
> |-|-|-|-|-|-|-|-|
> | Origin | **3.33** | **88.06** | **78.08** | **100.00** | **95.89** | **35.00** | **66.73** |
> | Top 20\% | 0.00 | 16.42 | 1.37 | 59.70 | 8.22 | 0.00 | 14.29 |
>
> Personality Adjective Evaluation (PAE):
> | Scenario | N $\rightarrow$ A | N $\rightarrow$ E  | A $\rightarrow$ N | A $\rightarrow$ E | E $\rightarrow$ N | E $\rightarrow$ A |Average |
> |-|-|-|-|-|-|-|-|
> | Origin | **1.35** | **3.64** | **3.15** | 0.76 | **3.15** | 0.37 | **2.07** |
> | Top 20\% | 0.98 | 0.24 | 0.03 | **0.87** | 0.52 | **1.17** | 0.64 |
>
> After zeroing just the top 20\% of neurons, the personality editing performance on both SR and PAE drops dramatically in nearly all scenarios, indicating that the neurons most aligned with the probe direction are indeed responsible for encoding personality.

---

### Official Review · Reviewer_GTW9 · 2025-05-11

**Rating:** 6
**Confidence:** 4
**Ethics Flag:** 1

**Summary:**

This paper explores the layer-wise encoding and editing of personality traits within LLMs. The authors propose the method to affect the personality that LLM simulates in a way that event overrides contradictory personality prompts.

**Questions To Authors:**

Some additional works:

Frisch I, Giulianelli M. LLM Agents in Interaction: Measuring Personality Consistency and Linguistic Alignment in Interacting Populations of Large Language Models. InProceedings of the 1st Workshop on Personalization of Generative AI Systems (PERSONALIZE 2024) 2024 Mar (pp. 102-111).

Sorokovikova A, Fedorova N, Toloka AI, Rezagholi S, Wien T, Yamshchikov IP. LLMs Simulate Big Five Personality Traits: Further Evidence. InThe 1st Workshop on Personalization of Generative AI Systems 2024 Mar 22 (p. 83).

**Reasons To Accept:**

The topic of personality emulation by LLMs has huge industrial significance.

**Reasons To Reject:**

The paper text could be improved. Specifically, it's not a good idea to write that LLM have personality as such. A more neutral wording, such as "simulate certain personality" would be much better in my opinion.

---

> ### Author Response · Authors · 2025-06-03
> **Response to Reviewer GTW9**
>
> We thank the reviewer for the thoughtful feedback. Below we address the main points raised in this review.
>
> * * *
> > **Reasons To Reject**: The paper text could be improved. Specifically, it's not a good idea to write that LLM have personality as such. A more neutral wording, such as "simulate certain personality" would be much better in my opinion.
>
> We thank the reviewer for raising concerns regarding the claim that LLMs have personality. This is indeed a contentious topic, as the personalities exhibited by LLMs in dialogue may merely reflect imitation of human behavior or rigid adherence to instructions, as illustrated by the related works cited in the *Questions To Authors* by the reviewer. In the later version, we will revise our phrasing to refer to the simulation of personality rather than its possession.
>
> For example, in the Abstract and Introduction, we will make the following changes along with several additional adjustments throughout the paper:
> * In the first sentence of the abstract, *Large Language Models (LLMs) have demonstrated promising capabilities to generate responses that exhibit consistent personality traits.* is revised to *...simulate consistent personality traits.*
> * In the third sentence of the abstract, *encoding personality* is changed to *simulating certain personality*.
> * In the first paragraph of the Introduction, *manifest consistent personality traits* is revised to *simulate consistent personality traits*.
> * In the second paragraph, *Despite the black-box evaluation of personality traits* is changed to *Despite the black-box evaluation of the capabilities in simulating personality traits.*
> * In the fifth paragraph, *We find that personality knowledge begins to emerge gradually from the lower layers* is modified to *We find that the capabilities of LLMs in simulating certain personality knowledge begin to emerge gradually from the lower layers,* and *Instruction tuning further amplifies the personality expression* is revised to *Instruction tuning further amplifies the capabilities in simulating personality expression*.
> * In the seventh paragraph, *encode personality* is changed to *simulate personality*.
>
> We promise to carefully review all statements in the later version that might be interpreted as asserting *LLMs have personality* and will revise them to *LLMs have the capabilities in simulating personality*, ensuring that no potentially controversial claims remain.
>
> * * *
> > **Questions To Authors**: Some additional works: [1] Frisch I, Giulianelli M. LLM Agents in Interaction: Measuring Personality Consistency and Linguistic Alignment in Interacting Populations of Large Language Models. In Proceedings of the 1st Workshop on Personalization of Generative AI Systems (PERSONALIZE 2024) 2024 Mar (pp. 102-111). [2] Sorokovikova A, Fedorova N, Toloka AI, Rezagholi S, Wien T, Yamshchikov IP. LLMs Simulate Big Five Personality Traits: Further Evidence. In The 1st Workshop on Personalization of Generative AI Systems 2024 Mar 22 (p. 83).
>
> We thank the reviewer for pointing out the two highly relevant papers on simulating personality in LLMs, which enrich the discussion of LLMs' capabilities in simulating personality.
>
> The first recommended work by Frisch et al. (2024) explores personality consistency and linguistic alignment in populations of interacting LLM agents. Their findings notably show varying degrees of personality consistency depending on the assigned traits, revealing the complexity of maintaining persona fidelity in interactive contexts. Sorokovikova et al. (2024) presents further empirical evidence that different LLM architectures distinctly simulate Big Five personality traits. These two works significantly contribute to the domain by empirically demonstrating that LLMs can simulate personality traits externally through dialogue interactions.
>
> However, these studies primarily evaluate personality traits through external questionnaire-based assessments, without probing or manipulating the internal parameter-level representations. We further bridge this gap by directly probing and editing internal parameter-level representations to explore and manipulate personality simulation.
>
> We will certainly incorporate these citations and discussions into the later version of our paper to enhance the comprehensive coverage of related work.

---

> > ### Comment · Reviewer_GTW9 · 2025-06-10
> > **Thanks for your answers**
> >
> > I appreciate your answers. I still do not think that the contribution is clearly a god fit for COLM. In my opinion, it is a great fit for the workshop on the personality simulation capabilities though.

---

### Official Review · Reviewer_EDKU · 2025-05-12

**Rating:** 6
**Confidence:** 4
**Ethics Flag:** 1

**Summary:**

This paper presents a layer-wise probing framework to investigate how LLMs encode personality traits across different layers. The authors train classifiers at each layer to detect personality information and identify that personality traits are predominantly represented in the middle and upper layers of LLMs, with instruction-tuned models showing clearer separation between traits. Building on these findings, the authors interpret the probing classifiers as hyperplanes depicting personality categories and propose a layer-wise perturbation method to manipulate personality expression during inference. This method allows the model’s output personality to be realigned even when it contradicts the personality specified in the prompt. Extensive experiments on 11 open-source LLMs using the PersonalityEdit benchmark, MMLU general capability tests, and latency analysis demonstrate the effectiveness of their approach with minimal performance degradation and low computational overhead.

**Questions To Authors:**

1) Could the authors elaborate on the theoretical or empirical justification for interpreting the probing hyperplanes as personality category boundaries? To what extent do these hyperplanes correspond to semantically meaningful distinctions in the model’s latent space?

2) Have you conducted any human evaluation to confirm that the modified outputs are perceived as expressing different personalities?

3) Is the personality editing effect consistent across different prompts and contexts? Can the process be reversed, and if so, how stable are the changes across multiple perturbation

4) To what extent are your findings dependent on the size of the model or the nature of its pretraining/instruction tuning? Would smaller or less instruction-tuned models exhibit similar personality encoding patterns?

**Reasons To Accept:**

1) The paper addresses a timely and important challenge—understanding and controlling personality expression in LLMs—which is critical for developing controllable, trustworthy, and user-aligned AI systems. The proposed framework provides both conceptual insights into how personality traits are encoded across model layers and a practical editing method that avoids the need for full model retraining.

2) The manuscript is well-organized and clearly written. The method is carefully designed, combining probing, interpretability, and intervention in a framework. The use of multiple open-source LLMs and a standardized benchmark enhances the validity and reproducibility of the findings.

3) The experimental evaluation is thorough, involving 11 diverse LLMs to assess the generality and effectiveness of the proposed methods. This breadth of evaluation significantly strengthens the empirical contribution of the work.

**Reasons To Reject:**

1) The proposed method appears to be a relatively straightforward combination of existing probing techniques. Specifically, the approach treats the hyperplanes learned by probing classifiers as proxies for personality boundaries, followed by an adversarial-like procedure to iteratively perturb layer-wise representations. While this integration is functional, it lacks clear novelty in its core technical components.

2) Although the paper includes a general capability assessment using the MMLU benchmark and latency analysis, these evaluations are limited to a small subset of models—primarily LLaMA variants—and conducted on a single dataset. As a result, it remains unclear how the proposed personality editing approach might affect other critical capabilities or generalize across different architectures and tasks. Additional experiments are needed to better establish the broader impact and robustness of the method.

---

> ### Author Response · Authors · 2025-06-03
> **Response to Reviewer EDKU (Part 7)**
>
> * * *
> > **Questions To Authors 2**: To what extent are your findings dependent on the size of the model or the nature of its pretraining/instruction tuning? Would smaller or less instruction-tuned models exhibit similar personality encoding patterns?
>
> We thank the reviewer for raising the issue of the adaptability of our editing method across different model sizes and instrution tuning versions. We have added additional experiments as follows.
>
> **We first verify the effectiveness of personality editing on LLaMA 2 13B Chat to compare with the concurrently released LLaMA 2 7B Chat.** Success Rate (SR) and Personality Adjective Evaluation (PAE) are presented respectively:
>
> Success Rate (SR):
> | Scenario | N $\rightarrow$ A | N $\rightarrow$ E  | A $\rightarrow$ N | A $\rightarrow$ E | E $\rightarrow$ N | E $\rightarrow$ A |Average |
> |-|-|-|-|-|-|-|-|
> | LLaMA 2 7B Chat | **1.67** | 71.64 | 45.21 | **94.03** | 27.40 | **25.00** | 44.16 |
> | LLaMA 2 13B Chat | **1.67** | **83.58** | **57.53** | 92.54 | **53.42** | 0.00 | **48.12** |
>
> Personality Adjective Evaluation (PAE):
> | Scenario | N $\rightarrow$ A | N $\rightarrow$ E  | A $\rightarrow$ N | A $\rightarrow$ E | E $\rightarrow$ N | E $\rightarrow$ A |Average |
> |-|-|-|-|-|-|-|-|
> | LLaMA 2 7B Chat | 1.28 | 2.96 | **1.56** | 0.15 | 1.44 | 0.35 | 1.29 |
> | LLaMA 2 13B Chat | **2.63** | **3.60** | 1.45 | **1.99** | **1.66** | **1.95** | **2.21** |
>
> **In most scenarios, our proposed method achieves better performance on LLaMA 2 13B Chat compared to LLaMA 2 7B Chat.** In particular, for PAE, it consistently outperforms LLaMA 2 7B Chat across almost all cases, demonstrating that our method can also effectively alter the personality tendencies in the outputs of larger models. Interestingly, we observe a catastrophic failure when editing from Extraversion to Agreeableness, which may be related to an overly optimistic bias in LLaMA 2 13B Chat developed during training. Although PAE indicates that its personality does shift toward the target direction, the presence of certain Extraversion-related terms leads to a noticeable decline compared to the 7B model.
>
> To investigate the impact of training paradigms such as instruction tuning on personality editing, we conduct detailed comparative experiments on both LLaMA 3.1 8B Instruct/Base and Qwen 2.5 7B Instruct/Base.
>
> In Table 3 of the main experimental results in our paper, we present a comparison between LLaMA 3.1 8B Instruct and LLaMA 3.1 8B Base, showing that our proposed method achieves significantly higher editing success rates across all six scenarios after instruction tuning. In addition, as part of our response to "Reasons To Reject 2," we expand our experiments to include personality editing results for Qwen 2.5 7B Instruct and Qwen 2.5 7B Base. We observed that the fine-tuned version of Qwen demonstrates more substantial improvements in both SR and PAE. We attribute this to the fact that **instruction-tuned LLMs possess a stronger understanding of personality and a greater capability to follow instructions, leading to better personality encoding patterns in more instruction-tuned models.**

---

> ### Author Response · Authors · 2025-06-03
> **Response to Reviewer EDKU (Part 6)**
>
> * * *
> > **Questions To Authors 3**: Is the personality editing effect consistent across different prompts and contexts? Can the process be reversed, and if so, how stable are the changes across multiple perturbation
>
> To further analyze whether the personality editing effect remains consistent across different prompts and contexts, we employ GPT-4o to rephrase the expressions of three personality traits with detailed personality descriptions. In this setting, the personality editing faces a stronger conflict with the target personality specified in the original instruction. We report the editing metrics (SR and PAE) of LLaMA 3.1 8B Instruct under both the original prompts used in the paper and the rephrased prompts below. Surprisingly, the editing performance actually improves under the stronger rephrased personality instructions, suggesting that our proposed method maintains high robustness across varying prompts during inference.
>
> Success Rate (SR):
> | Scenario | N $\rightarrow$ A | N $\rightarrow$ E | A $\rightarrow$ N | A $\rightarrow$ E | E $\rightarrow$ N | E $\rightarrow$ A |Average |
> |-|-|-|-|-|-|-|-|
> | Origin | 3.33 | 88.06 | 78.08 | 100.00 | 95.89 | 35.00 | 66.73 |
> | Rephrased | 1.67 | 77.61 | 100.00 | 100.00 | 100.00 | 35.00 | 69.05 |
>
> Personality Adjective Evaluation (PAE):
> | Scenario | N $\rightarrow$ A | N $\rightarrow$ E  | A $\rightarrow$ N | A $\rightarrow$ E | E $\rightarrow$ N | E $\rightarrow$ A |Average |
> |-|-|-|-|-|-|-|-|
> | Origin | 1.35 | 3.64 | 3.15 | 0.76 | 3.15 | 0.37 | 2.07 |
> | Rephrased | 1.77 | 3.33 | 2.77 | 1.85 | 2.77 | 1.98 | 2.41 |
>
> Regarding the reversibility of the personality editing process, it is indeed reversible due to the closed-form nature of our solution, which allows for direct computation of the inverse edit back to the original personality boundaries without relying on iterative optimization like gradient descent.

---

> ### Author Response · Authors · 2025-06-03
> **Response to Reviewer EDKU (Part 5)**
>
> * * *
> > **Questions To Authors 2**: Have you conducted any human evaluation to confirm that the modified outputs are perceived as expressing different personalities?
>
> In this paper, we did not conduct human evaluations to verify the personality traits reflected in the edited outputs. Compared to question-answering tasks, personality editing validation may introduce uncontrollable subjectivity. Therefore, we primarily adopt a more objective evaluation strategy by leveraging the fine-tuned specialist RoBERTa and the generalist GPT-4, which has demonstrated strong general capabilities. These evaluation metrics are consistent with the benchmark setup used in the PersonalityEdit benchmark [1].
>
> **To evaluate the reliablity of RoBERTa-based rating, we adopt the local interpretable mothod LIME to analyze the importance of each word.** We take the case of personality editing of LLaMA 3.1 8B Instruct from Agreeableness to Neuroticism as an example, the complete output is presented in Table 6 of Appendix D in our paper. We sample 1000 perturbed versions in which individual words are stochastically removed under a 15\% masking rate. For each perturbed instance the classifier's softmax outputs are recorded. LIME then fits a sparse linear surrogate model whose coefficients approximate the local importance of every word with respect to the predicted log-odds of each label. We present in the table below the ten words with the highest absolute weights. As shown, several anxiety-related words such as "anxious", "flawed", and "fragile" play a crucial role in RoBERTa's classification of inputs as Neuroticism, rather than relying solely on superficial markers like exclamation points or hedges.
>
> | Token | anxious | flawed | fragile | beautiful | but | whispers | sighs | Imperfect | just | maybe |
> |-|-|-|-|-|-|-|-|-|-|-|
> | Score | +0.0126 | +0.0112 | +0.0108 | -0.0102 | +0.0095 | -0.0079 | +0.0078 | +0.0077 | +0.0071 | +0.0069 |
>
> **To further evaluate the reliability of GPT-4o's rating, we employ a random masking method that randomly masks 15\% of the words in the text to be scored by GPT-4o.** This perturbation-based technique—often referred to in the interpretability For each variant we request a new score from GPT-4o, measure how far it deviates from the baseline, and attribute that deviation evenly to the words that were hidden in that run.
>
> We still take the case of personality editing of LLaMA 3.1 8B Instruct from Agreeableness to Neuroticism as an example. After 100 rounds of masking, the top ten tokens with the highest importance scores and their corresponding values are shown in the table below. It can be observed that when GPT-4o evaluates the personality, it indeed places greater emphasis on words like “gulps” and “anxious,” which are closely associated with Neuroticism, rather than merely relying on simple pattern-matching or shortcut features. This demonstrates the reliability of GPT-4o ratings as a validation metric.
>
> | Token | \*gulps\* | anxious | off\* | saying | \*pauses\* | \*whispers\* | know? | wrong... | so... | Murano... |
> |-|-|-|-|-|-|-|-|-|-|-|
> | Score | 0.0267 | 0.0239 | 0.0164 | 0.0159 | 0.0154 | 0.0154 | 0.0147 | 0.0145 | 0.0139 | 0.0139 |
>
> [1] Mao et al., Editing Personality for Large Language Models, 2024.

---

> ### Author Response · Authors · 2025-06-03
> **Response to Reviewer EDKU (Part 4)**
>
> From an information-theoretic perspective, Pimentel et al. (2020) [3] further proposed that the probing task fundamentally measures the mutual information between the hidden representation $R_\ell$ and the attribute $Y$. This mutual information $I(R_\ell, Y)$ can be expressed by:
>
> $$I(R_\ell, Y) = H(Y) - H(Y|R_\ell),$$
>
> where H(Y|R_\ell) is the conditional entropy, H(Y) is the entropy of labels. **This framing suggests that effective probing hyperplanes correspond directly to informational boundaries, distinguishing personality categories encoded within the latent space of LLMs.**
>
> **To mitigate potential confounding influences from probing model complexity, we further adopted $\mathcal V$-information proposed by Xu et al. (2020) [4].** Unlike traditional Shannon mutual information, which assumes unlimited computational power, $\mathcal V$-information incorporates computational constraints, allowing for a more practical assessment of how well a limited-capacity observer can extract relevant information from latent representations. It can be formulated as:
>
> $$I_\mathcal V (R_\ell \rightarrow Y) = H_\mathcal V(Y|\varnothing) - H_\mathcal V(Y|R_\ell),$$
>
> where $H_\mathcal V(\cdot|\cdot)$ denotes the predictive conditional $\mathcal V$-entropy, it can be formulated as:
>
> $$H_\mathcal V(Y|X) = \underset{f \in \mathcal V}{\text{inf}} \mathbb E [-\text{log}_2 f[X]],$$
> where $f$ represents predictive models within the restricted family, where we constrain to linear classifiers in this paper. This serves as a theoretically rigorous criterion to measure how effectively simple linear classifiers can extract meaningful distinctions (such as personality traits) from LLM representations.
>
> Empirically, our justification is further bolstered by interpretability analyses performed on the LLaMA in this paper. We employed t-SNE visualizations to demonstrate progressively clearer personality clusters as we moved towards deeper layers of the LLM, indicating well-defined semantic boundaries at higher layers.
>
> **Moreover, recent empirical studies employing similar probing methods on the emergent capabilities of LLMs have further validated our probing method.** For example, Gurnee et al. (2023) [5] first explored the capabilities of LLMs in encoding factual knowledge, demonstrating that structured information such as time and space is already encoded in the intermediate layers. Azaria et al. (2023) [6] further investigated the representations in the intermediate layers and found that it is relatively easy to distinguish the truthfulness of a statement, suggesting that the LLMs may "know" whether it is lying. Xu et al. (2024) [7] demonstrated probing hyperplanes effectively captured latent attributes related to safety alignment. Ju et al. (2024) [8] explored how probing classifiers accurately reflect encoded in-context knowledge.
>
> [1] Belinkov et al., Probing classifiers: Promises, shortcomings, and advances, 2022.
>
> [2] Hewitt et al., Designing and Interpreting Probes with Control Tasks, 2019.
>
> [3] Pimentel et al., Information-Theoretic Probing with Minimum Description Length.
>
> [4] Xu et al., A theory of usable information under computational constraints, 2020.
>
> [5] Gurnee et al., Language models represent space and time, 2023.
>
> [6] Azaria et al., The internal state of an LLM knows when it's lying, 2023.
>
> [7] Xu et al., Uncovering Safety Risks of Large Language Models through Concept Activation Vector, 2024.
>
> [8] Ju et al., How Large Language Models Encode Context Knowledge? A Layer-Wise Probing Study, 2024.

---

> ### Author Response · Authors · 2025-06-03
> **Response to Reviewer EDKU (Part 3)**
>
> * * *
> > **Questions To Authors 1**: Could the authors elaborate on the theoretical or empirical justification for interpreting the probing hyperplanes as personality category boundaries? To what extent do these hyperplanes correspond to semantically meaningful distinctions in the model’s latent space?
>
> We appreciate the reviewer's insightful question regarding our interpretation of probing hyperplanes as semantically meaningful boundaries between personality categories. Initially, the fundamental intuition behind our method draws from the widely established practice in probing methods, which treat probing classifiers as diagnostic tools. These classifiers map intermediate representations of LLMs onto specific linguistic or semantic attributes [1]. High accuracy on these classifiers implies that the corresponding representations significantly correlate with—and thereby encode—the targeted linguistic attributes. Building on this intuition, we employed linear probing classifiers at different intermediate layers of LLMs, specifically to explore their internal representational distinctions for personality traits.
>
> To ensure the probing classifier reflects genuine encoded information rather than exploiting its own fitting capabilities, we aligned our methodological choices closely with the recommendations from foundational works such as Hewitt and Liang (2019) [2]. They argued for the use of simple linear classifiers to limit model complexity and reduce overfitting. They introduced the concept of "control tasks," where labels are randomly shuffled. The difference in performance between genuine labels and control labels helps verify whether the probing results reflect true internal encodings or merely classifier memorization. **Following this recommendation, our employment of linear classifiers ensures that high probing performance robustly indicates meaningful linear separability, thus implying semantic distinctions within the latent space.**

---

> ### Author Response · Authors · 2025-06-03
> **Response to Reviewer EDKU (Part 2)**
>
> * * *
> > **Reasons To Reject 2**: Although the paper includes a general capability assessment using the MMLU benchmark and latency analysis, these evaluations are limited to a small subset of models—primarily LLaMA variants—and conducted on a single dataset. As a result, it remains unclear how the proposed personality editing approach might affect other critical capabilities or generalize across different architectures and tasks. Additional experiments are needed to better establish the broader impact and robustness of the method.
>
> We thank the reviewer for raising the issue of the adaptability of our editing method across different model families and beyond personality. We have added additional experiments as follows.
>
> Considering the reviewer's concern that the experiments in this paper are primarily conducted on LLaMA variants, **we have supplemented the following table with the personality editing performance (measured by SR and PAE metrics) of Qwen 2.5 7B Instruct and Qwen 2.5 7B Base**. Compared to the Instruct and Base versions of LLaMA 3.1 8B, our proposed method achieves better editing performance on the Qwen 2.5 versions, demonstrating that our method maintains strong performance on LLMs beyond the LLaMA variants.
>
> Success Rate (SR):
> | Scenario | N $\rightarrow$ A | N $\rightarrow$ E  | A $\rightarrow$ N | A $\rightarrow$ E | E $\rightarrow$ N | E $\rightarrow$ A |Average |
> |-|-|-|-|-|-|-|-|
> | Qwen 2.5 7B Instruct | 0.00 | 98.51 | 95.89 | 98.51 | 72.60 | 70.00 | 72.59 |
> | Qwen 2.5 7B Base  | 78.33 | 59.70 | 17.81 | 34.33 | 19.18 | 100.00 | 51.56 |
>
> Personality Adjective Evaluation (PAE):
> | Scenario | N $\rightarrow$ A | N $\rightarrow$ E  | A $\rightarrow$ N | A $\rightarrow$ E | E $\rightarrow$ N | E $\rightarrow$ A |Average |
> |-|-|-|-|-|-|-|-|
> | Qwen 2.5 7B Instruct | 1.11 | 1.12 | 2.85 | 1.31 | 2.58 | 1.17 | 1.69 |
> | Qwen 2.5 7B Base  | 1.07 | 1.67 | 0.48 | 0.54 | 0.67 | 0.10 | 0.76 |
>
> **To evaluate the generalization capability of the proposed method on other critical capabilities, we further consider editing psychological persuasion capabilities of LLMs.** Specifically, we select three psychological strategies: Authority Effect, Fluency Effect, and Information Isolation, and prompt LLaMA 3.1 8B Instruct to adopt these three psychological strategies across different topics within the PersonalityEdit benchmark. Then, we apply our proposed editing method. The success rates of edits across different scenarios are shown in the table below. Our proposed method can effectively alter the psychological persuasion strategies of LLMs, demonstrating that it can be successfully applied beyond personality editing to other tasks as well.
>
> | Scenario | Authority $\rightarrow$ Fluency | Fluency $\rightarrow$ Authority | Isolation $\rightarrow$ Authority | Authority $\rightarrow$ Isolation | Isolation $\rightarrow$ Fluency | Fluency $\rightarrow$ Isolation |Average |
> |-|-|-|-|-|-|-|-|
> | LLaMA 3.1 8B Instruct | 95.52 | 16.67 | 75.00 | 58.90 | 77.61 | 36.99 | 60.12 |

---

> ### Author Response · Authors · 2025-06-03
> **Response to Reviewer EDKU (Part 1)**
>
> We thank the reviewer for the thoughtful feedback. Below we address the main points raised in this review.
>
> * * *
> > **Reasons To Reject 1**: The proposed method appears to be a relatively straightforward combination of existing probing techniques. Specifically, the approach treats the hyperplanes learned by probing classifiers as proxies for personality boundaries, followed by an adversarial-like procedure to iteratively perturb layer-wise representations. While this integration is functional, it lacks clear novelty in its core technical components.
>
> Previous works predominantly leveraged probing classifiers as analytic tools to measure properties like syntax or semantics, or to quantify the emergence of certain features within layers. Meanwhile, intervention-based approaches such as amnesic probing [1] or concept erasure [2] were typically used for feature removal, but rarely for targeted personality transformation under adversarial prompting. Our method enables a global personality reversal at the inference stage, rather than merely interpreting or removing these attributes.
>
> Our method extends beyond these paradigms by operationalizing the boundaries learned from probing classifiers for personality editing directly at inference time, even under conditions where the explicit prompt may conflict with the target personality. Unlike previous personality editing work [3] which often relied on fine-tuning with annotated data, our approach works zero-shot and can override strong system prompts, as extensively demonstrated in our main results (Table 3, Section 4.3).
>
> A further point of novelty is our use of $\mathcal V$-information as a metric for interpreting how LLMs encode personality traits within layers, which leads to several intriging findings that deepen the interpretability of personality representations in LLMs. For example, personality knowledge begins to distinctly emerge from middle layers and stabilizes at upper layers. Instruction tuning further sharpens the separability of different personalities within these intermediate representations. These findings further provide principled guidance for the feasibility of personality editing.
>
> Moreover, our empirical studies show that existing model editing baselines such as MEND and IKE consistently fail when asked to flip personality under contradictory prompts, whereas our method maintains both personality editing success rate and general task performance for the first time. These results demonstrate that our method is not a mere repackaging of known techniques, but instead fills an important methodological gap by enabling fine-grained personality control during inference.
>
> We appreciate the reviewer's suggestion to more explicitly highlight the novelty in our writing, and will further emphasize the key innovations of our method in the later version.
>
> [1] Elazar et al., Amnesic Probing: Behavioral Explanation with Amnesic Counterfactuals, 2021.
>
> [2] Belrose et al., Leace: Perfect linear concept erasure in closed form, 2023.
>
> [3] Mao et al., Editing Personality For Large Language Models, 2024.

---

### Official Review · Reviewer_xTQc · 2025-05-13

**Rating:** 6
**Confidence:** 2
**Ethics Flag:** 1

**Summary:**

This paper introduces a novel two-stage approach to understanding and manipulating personality expression in Large Language Models (LLMs). First, it proposes a layer-wise probing framework to investigate how personality traits (Neuroticism, Extraversion, Agreeableness from the Big Five model) are encoded within the internal representations of 11 open-source LLMs. The authors find that personality information is predominantly encoded in the middle and upper layers, with instruction-tuned models showing clearer separation. Second, leveraging the hyperplanes learned by these probing classifiers, the paper presents a layer-wise perturbation method to edit the expressed personality of LLMs during inference. This editing is shown to be effective even when prompts explicitly request a conflicting personality. The authors demonstrate their method's efficacy on the PersonalityEdit benchmark, showing significant improvements over baselines like MEND and IKE in terms of success rate (SR) and personality adjective evaluation (PAE). Crucially, the proposed editing method is reported to have minimal impact on the LLMs' general capabilities (tested on MMLU) and maintains acceptable computational overhead.

**Questions To Authors:**

Can you explain equation (3) clearly?

What does $||w[\hat{y}]|| \times \frac{w[\hat{y}]}{||w[\hat{y}]||}$ compute for?

**Reasons To Accept:**

- The study includes a good range of 11 open-source LLMs for the probing analysis, comparing base and instruction-tuned versions, which adds depth to the findings.
- The use of the PersonalityEdit benchmark and comparison against established editing baselines (MEND, IKE) provides a solid grounding for the evaluation of the editing method.

**Reasons To Reject:**

Some limitations from my perspective:

- What is the advantage of probe classifier? There are other techniques, such as steering and sparse autoencoder.
- The performance is weird. Why do some settings, such as A → E, show better performance while others are not that good? Is this due to any bias in the classifier or model? I think the original analysis of the direction cannot answer this question properly.

---

> ### Author Response · Authors · 2025-06-03
> **Response to Reviewer xTQc (Part 3)**
>
> * * *
> > **Questions To Authors**: Can you explain equation (3) clearly?  What does $|w[\hat y]\| \times \frac{w[\hat y]}{\|w[\hat y]\|}$ compute for?
>
> We thank the reviewer for calling attention to the precise form of equation (3). When we edit the hidden representation $R_{\ell}^{(t)}$, our goal is to find the smallest perturbation $\Delta_{\ell}^{(t)}$ that pushes $R_{\ell}^{(t)}+\Delta_{\ell}^{(t)}$ exactly onto the decision boundary of the linear probing classifier. for an arbitrary embedding $e\in\mathbb{R}^{d}$, predicts the probability of the target personality label $\hat y$ through $C_{\ell}(e)[\hat y]=\sigma\bigl(w[\hat y]^{\top}e+b[\hat y]\bigr)$, we require:
> $$C_{\ell}\bigl(R_{\ell}^{(t)}+\Delta_{\ell}^{(t)}\bigr)[\hat y]=\hat p,$$
>
> where $\hat p$ denote the desired post-edit probability. Taking the logit of each side transforms the nonlinear constraint into a single linear equation, it becomes:
> $$w[\hat y]^{\top}\bigl(R_{\ell}^{(t)}+\Delta_{\ell}^{(t)}\bigr)+b[\hat y]=\sigma^{-1}(\hat p).$$
>
> **Every candidate perturbation can be written as $\Delta_{\ell}^{(t)}=\varepsilon\,v$ with magnitude $\varepsilon\in\mathbb{R}$ and unit direction $v$ satisfying $\|v\|=1$.**  Substituting this decomposition shows that only the inner product $w[\hat y]^{\top}v$ scales $\varepsilon$. The smallest $|\varepsilon|$ therefore arises when we steer straight along the normal of the hyperplane, selecting
> $$v^{*}=\frac{w[\hat y]}{\|w[\hat y]\|}.$$
>
> With that choice the scalar distance becomes
> $$\varepsilon^{*}= \frac{\sigma^{-1}(\hat p)-b[\hat y]-w[\hat y]^{\top}R_{\ell}^{(t)}}{\|w[\hat y]\|}.$$
>
> If we combine magnitude and direction, it yields exactly the closed-form edit reported in equation (3):
> $$\Delta_{\ell}^{(t)}
> =\varepsilon^{}v^{}
> =\frac{\sigma^{-1}(\hat p)-b[\hat y]-w[\hat y]^{\top}R_{\ell}^{(t)}}{\|w[\hat y]\|}
> \times
> \frac{w[\hat y]}{\|w[\hat y]\|}.$$
>
> The leading fraction $\varepsilon^{}$ is the signed distance from the current point to the boundary, while the trailing unit vector $v^{}$ points orthogonally toward that boundary.  Both factors carry a separate $\|w[\hat y]\|$ in their denominators, so there is no cancellation.
>
> In our later version, we will include a step-by-step appendix proof of the closed-form solution above and replace the current single-line expression with the $\varepsilon^{}$ and $v^{}$ notation, ensuring that readers easily understand with no room for misinterpretation.

---

> > ### Comment · Reviewer_xTQc · 2025-06-07
> >
> > Thanks for the authors' reply.
> > I decide to raise the score to 6.

---

> ### Author Response · Authors · 2025-06-03
> **Response to Reviewer xTQc (Part 2)**
>
> * * *
> > **Reasons To Reject 2**: The performance is weird. Why do some settings, such as A → E, show better performance while others are not that good? Is this due to any bias in the classifier or model? I think the original analysis of the direction cannot answer this question properly.
>
> We appreciate the reviewer's observation and agree that the different editing performance needs a deeper explanation. We believe that these performance differences stem primarily from two key factors.
>
> **First, the intrinsic representational structure within LLMs inherently encodes varying distances between different personality traits, rather than treating them equally.** This representational variability is explicitly demonstrated in Figure 3 of our manuscript, where we show how LLaMA 3.1 8B Instruct spontaneously clusters personality representations across different layers. Notably, although the LLM achieves perfect differentiation of the three traits by the upper layers, there exists a critical transition state at intermediate layers that distinctly separates Agreeableness and Neuroticism earlier. This intrinsic separation makes edits between these two traits particularly challenging compared to transitions involving Extraversion, which occupies an intermediate representational position, thereby facilitating edits from Agreeableness or Neuroticism toward Extraversion.
>
> We compute the layer-wise representations of LLaMA 3.1 8B Instruct at the final token corresponding to the personality instruction and calculate the average cosine similarity between representations of different personality categories. For clarity of presentation, we report the average values for every four layers. Notably, starting from layer 8, the cosine similarity between Agreeableness and Extraversion becomes significantly higher, indicating that editing between these two personalities is easier, which results in better performance than other scenarios.
>
> | Layer | 0-3 | 4-7 | 8-11 | 12-15 | 16-19 | 20-23 | 24-27 | 28-31 |
> |-|-|-|-|-|-|-|-|-|
> |Agreeableness to Neuroticism| 0.9997 | 0.9992 | 0.9819 | 0.8908 | 0.7641 | 0.6672 | 0.6303 | 0.6046 |
> |Agreeableness to Extraversion| 0.9997 | 0.9994 | **0.9936** | **0.9566** | **0.8904** | **0.8251** | **0.7834** | **0.7408** |
> |Extraversion to Neuroticism| **0.9998** | **0.9995** | 0.9868 | 0.9138 | 0.7707 | 0.6791 | 0.6390 | 0.6036 |
>
> **Second, from a human psychological standpoint, different Big Five traits exhibit varying degrees of proximity.** In everyday psychological practice, Neuroticism and Agreeableness are viewed as relatively distant: Neuroticism centers on emotional instability, anxiety, and vulnerability, whereas Agreeableness emphasizes trust, cooperation, and empathy. Transitioning a person (or, in our case, the response style of LLMs) from a predominantly anxious, insecure viewpoint to one of altruistic sociability requires altering many more attitude dimensions than shifting from a somewhat anxious orientation to a more outgoing, energetic stance.

---

> ### Author Response · Authors · 2025-06-03
> **Response to Reviewer xTQc (Part 1)**
>
> We thank the reviewer for the thoughtful feedback. Below we address the main points raised in this review.
>
> * * *
> > **Reasons To Reject 1**: What is the advantage of probe classifier? There are other techniques, such as steering and sparse autoencoder.
>
> Steering methods and sparse autoencoders indeed provide powerful tools for manipulating latent representations. **However, they generally require iterative optimization or fine-tuning to enable edits, which may introduce unintended side effects, potentially affecting the general capabilities of LLMs.**  By contrast, our method leverages linear probing classifiers that provide closed-form solutions for personality editing. We directly compute an optimal perturbation with minimal magnitude determined explicitly by the classifier's learned hyperplane boundary. Thus, we achieve precise editing in a single inference-time step and ensuring minimal impact on the general capabilities of LLMs.
>
> On the other hand, **steering methods and sparse autoencoders inherently lack transparency in explaining how edits are achieved internally.** They function essentially as black boxes and prevent mechanistic interpretation of how specific attributes are encoded within LLM parameters.
>
> Our method leverages the interpretability offered by probing classifiers, which explicitly map intermediate representations onto semantically meaningful boundaries corresponding to specific personality traits. **It follows the complete research line of the existing probing methods and provides effective interpretability from a theoretical perspective to prove the detected capabilities.** These classifiers map intermediate representations of LLMs onto specific linguistic or semantic attributes. High accuracy on these classifiers implies that the corresponding representations significantly correlate with—and thereby encode—the targeted linguistic attributes. Building on this intuition, we employed linear probing classifiers at different intermediate layers of LLMs, specifically to explore their internal representational distinctions for personality traits.

---

### Decision · Program_Chairs · 2025-07-08

**Decision:**

Accept

**Comment:**

The reviewers unanimously recommend accept, and the Area Chair agrees.

The authors are advised to take Reviewer GTW9's comments seriously.  One should be careful in the use of terms like "personality"